# Facing the Elephant in the Room: Visual Prompt Tuning or Full Finetuning?

**Cheng Han**[1,2]**, Qifan Wang**[3]**, Yiming Cui**[4]**, Wenguan Wang**[5]**, Lifu Huang**[6]**,
Siyuan Qi**[7]**, Dongfang Liu**[1]*

Rochester Institute of Technology[1], University of Missouri - Kansas City[2], Meta AI[3],
University of Florida[4], Zhejiang University[5], Virginia Tech[6], BIGAI[7]†

`{ch7858, dongfang.liu}@rit.edu, wqfcr@fb.com, cuiyiming@ufl.edu,`
`lifuh@vt.edu, wenguanwang.ai@gmail.com, syqi@bigai.ai`

## Abstract

As the scale of vision models continues to grow, the emergence of Visual Prompt Tuning (VPT) as a parameter-efficient transfer learning technique has gained attention due to its superior performance compared to traditional full-finetuning. However, the conditions favoring VPT (the "when") and the underlying rationale (the "why") remain unclear. In this paper, we conduct a comprehensive analysis across 19 distinct datasets and tasks. To understand the "when" aspect, we identify the scenarios where VPT proves favorable by two dimensions: task objectives and data distributions. We find that VPT is preferrable when there is 1) a substantial disparity between the original and the downstream task objectives (*e.g.*, transitioning from classification to counting), or 2) a similarity in data distributions between the two tasks (*e.g.*, both involve natural images). In exploring the "why" dimension, our results indicate VPT's success cannot be attributed solely to overfitting and optimization considerations. The unique way VPT preserves original features and adds parameters appears to be a pivotal factor. Our study provides insights into VPT's mechanisms, and offers guidance for its optimal utilization. The code is available at https://github.com/ChengHan111/VPT-or-FT.

## 1 Introduction

Recent advancements in artificial intelligence [38; 4; 93; 33], especially large models in natural language processing [113; 133; 58; 77; 131], have led to the development of large-scale vision models (*e.g.*, BiT [53], ViT [24], Swin [66], Florence [124]) that have revolutionized various tasks. These models are typically trained on extensive datasets (*e.g.*, ImageNet-21k [89], Open Images [55]) and then finetuned to adapt to specific downstream tasks [47] (*e.g.*, Coco-stuff [7], FGVC [49], VTAB-1k [127]). As models drastically scale up to boost performance, full finetuning that unfreezes all parameters for the update becomes computationally and storage-intensive [111; 14] and impractical. This traditional method can also lead to the loss of valuable knowledge acquired during pretraining, thereby limiting the models' generalizability [49; 134].

To address this issue, the search for parameter-efficient finetuning methods in vision is an ongoing endeavor. These parameter-efficient approaches freeze most parts of the pretrained model, and only tune the rest or insert customized learnable modules, which significantly reduce the number of learnable parameters compared to full finetuning [16; 48; 69; 85; 129]. Among all these methods, visual prompt tuning [49], which is again inspired by language-domain prompting works [68; 102; 65; 64; 34], stands out as one of the most prominent techniques in this field. VPT introduces a small number of extra trainable parameters (typically less than 1% of model parameters) in the input space of the transformer layers while keeping the backbone frozen. For the first time, it achieves superior performance over full finetuning in image recognition, and is quickly adapted into other visual tasks such as image segmentation [62; 120; 130; 110], image captioning [135], *etc*.

With VPT's growing popularity [3; 99; 112], a research question naturally arises: *when and why does visual prompt tuning (VPT) outperform full finetuning (FT) as transfer learning paradigm?*

---

*Corresponding author
†National Key Laboratory of General Artificial Intelligence, Beijing Institute for General Artificial Intelligence

To address this question, we conducted extensive experiments on 19 diverse datasets and tasks, wherein VPT outperformed FT in 16 instances. To discern **when** VPT is preferred, we categorized transfer learning scenarios into four groups based on the disparity between the original and downstream tasks, based on task objectives and data distributions (Figure 1). We found that VPT is advantageous when (*Case 1*) a substantial gap exists between the original and downstream tasks (*e.g.*, transitioning from classification to counting) or (*Case 2*) the data distributions are notably similar between the tasks (*e.g.*, both deal with natural images). The size of the downstream task dataset also influences this preference, with FT becoming increasingly favorable as the data size grows.

We further investigate **why** VPT excels. While one plausible hypothesis suggests that FT is more prone to overfitting due to its numerous tunable parameters, our experiments reveal that this is only part of the story. Overfitting is primarily observed in *Case 1* (high task disparity), while in *Case 2*, both methods show no trend for overfitting. A possible explanation is that the additional parameters introduced by VPT offer additional dimensions to escape the local minima of the pretrained model. However, empirical results do not support this assumption when we compare methods with additional dimensions. Our exploration of various tuning method variations underscores the importance of preserving the original feature for achieving superior performance, albeit through a non-obvious pathway. Our results suggest that VPT preserves features and add parameters in a unique manner that is pivotal in this process.

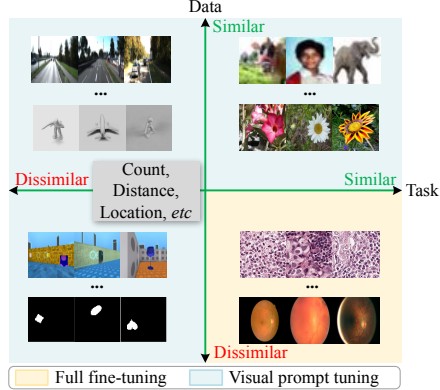

Figure 1: VPT is identified to be preferable in 3 out of 4 transfer learning scenarios when downstream data is limited.

Overall, this study aims to provide a thorough reassessment of the effectiveness of VPT compared to FT. The primary **contributions** of this paper can be summarized as follows:

- We identify the transfer learning scenarios where VPT proves advantageous by considering two critical dimensions: data distributions and task objectives. Notably, VPT is preferable in 3 out of 4 quadrants, with FT gradually closing the performance gap as the downstream data size increases.

- We uncover that overfitting alone does not account for all of VPT's success. Moreover, the advantage of VPT cannot be attributed solely to additional parameters aiding the model in escaping local minima. The unique introduction of extra parameters plays a key role in VPT's performance.

- To showcase the efficacy of prompt tuning over full finetuning, we provide attention map visualizations for both methods and demonstrate that visual prompts could enhance feature learning.

## 2 RELATED WORK

**Full finetuning.** The emergence of convolutional neural networks (CNNs) (*e.g.*, VGG [96], ResNet [40], MobileNet [45; 90; 44]) makes a significant contribution to the rapid advancements in computer vision. The trained CNN backbones (knowledge) on large-scale datasets can be easily transferred to adapt downstream vision tasks through finetuning (normally full finetuning), which is termed as "pretrain-then-finetune". This approach allows for a flexible adaptation, standing in contrast to earlier methods that relied on hard-designed task-specific models [19; 5; 20]. Instead of training each task from scratch, it initializes the model with pre-trained weights, updating all the parameters from the original backbone with separate instances for different tasks. Full finetuning is a common practice under this paradigm, which keeps being a powerful yet affordable approach [47; 134; 105; 13] with broad applicability. The subsequent prominent transformer-based architectures in vision (*e.g.*, ViT [24], Swin [66]) inherited the same pipeline for adaptation as training these backbones from scratch can hardly learn global features [24; 12] on small datasets (*i.e.*, Transformers extract features with less inductive bias, which contributes to continuously improve accuracy by increasing training data [26; 24]. However, the limited training data can not fully exploit the capacity). Recent breakthroughs in language for scaling up models in size led to stronger generality [4; 136; 83; 6]. Following this tendency, upcoming vision models (*e.g.*, Florence [124], CoCa [123]) are heavily parameterized, when following the common approach under "pretrain-then-finetune" paradigm, becomes impractical to fully finetune these models. This is primarily due to the

inherent parameter-inefficient nature and the storage-wise expensive requirements of full finetuning. Therefore, parameter-efficient finetuning methods become a critical research area.

**Visual Prompt Tuning.** With the significant growth in the scale of current vision models, the development of parameter-efficient finetuning methods under "pretrain-then-finetune" paradigm becomes increasingly critical. Current methods can be generally categorized into partial tuning [16; 48; 69], extra module [85; 129; 8] and prompt tuning [49; 50; 23; 118; 125], while partial tuning and extra module face several limitations that hinder their application: First, they generally cannot reach competitive results to full finetuning [49; 16; 48; 69]; Second, some require to insert specific architecture/block design [129; 85; 8] during tuning, rendering them non-universal solutions when considering various backbones. Prompt tuning [59], on the other hand, provides a straightforward solution with strong performance gains during finetuning, which is originally proposed for language-domain [68; 42; 63; 82]. The emergence of visual-related prompt tuning [36; 30; 115] has only recently come to the fore, signaling a new paradigm in parameter-efficient finetuning techniques in the field of computer vision. Generally, prompt tuning prepends extra sets of learnable parameters to the input sequence of backbones, and only updates these parameters during finetuning. While appearing to be simplistic, the paradigm of visual prompt tuning has exhibited satisfactory performance enhancements, as opposed to other parameter-efficient finetuning methods which require specific module designs (*i.e.*, extra module) [85; 129; 8] or coercive freezing of certain portions of the backbone network (*i.e.*, partial tuning) [16; 48; 69]. Moreover, these methods show a substantial performance gap to full finetuning, thereby rendering their widespread implementation under "pretrain-then-finetune" paradigm. Although current visual prompt tuning presents promising results, it remains unexplored in the comprehension of the underlying mechanisms that are responsible for their resounding success. In contrast, in the field of NLP, there are several works [13; 107] that investigate various aspects of prompt tuning (*e.g.*, data scale, evaluation, stability). In light of this view, this paper endeavors to explore and elucidate diverse facets pertaining to visual prompt tuning.

## 3 METHODOLOGY

**Full finetuning** is a widely used finetuning method under "pretrain-then-finetune" paradigm. In particular, given a pre-trained model with parameter denoted by $\theta$ and downstream dataset $D$, full finetuning aims to learn a new set of parameters $\theta'$ under the same model architecture as in pretraining. The objective of full finetuning is to minimize the corresponding loss function $L(\theta')$ between the predictions $f(x; \theta')$ and the ground truth $y$ by optimizing the parameters $\theta'$ over the entire network. While full finetuning obtains good performance with sufficient data, it is computationally expensive and requires the storage of all parameters.

**Visual Prompt Tuning**, in contrast, keeps the pretrained backbone model frozen while finetuning only a small set of task-specific visual prompts [49; 65; 34; 59; 119; 108], defined as $P = \{P_0, P_1, \ldots, P_{N-1}\}$, where $P_{i-1}$ is learnable vectors which are prepended to the input sequence of the $i$-$th$ layer (as shown in Figure 2(b)). The optimization process of prompt tuning involves updating only the newly added prompts while keeping the pretrained model $\theta$ fixed. As a consequence, prompt tuning is particularly beneficial in terms of limiting the overall parameter storage and updating, and preserving the learned knowledge without forgetting during the finetuning phase. In general, for a Vision Transformer [24] (ViT) with $N$ layers $L$, prompt tuning can be formulated as:

$$
\begin{aligned}
Z_1 &= L_1(P_0,\ E)\,, \\
Z_i &= L_i(P_{i-1},\ Z_{i-1}) \quad i = 2, 3, \ldots, N\,, \\
y &= \text{Head}(Z_N)
\end{aligned}
\tag{1}
$$

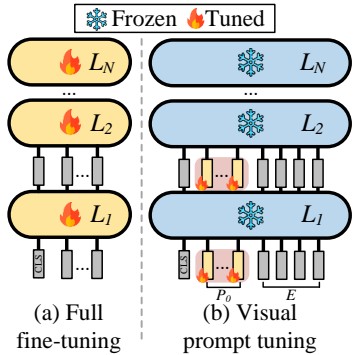

(a) Full fine-tuning    (b) Visual prompt tuning

Figure 2: **Full finetuning** *vs.* **visual prompt tuning**. Visual prompt tuning only learns a small set of prompts.

$Z_i$ is the contextual embeddings computed by the $i_{th}$ encoder layer. $E$ is the image patch embeddings initialized with frozen embedding projection from the backbone. A classification head is then used to map the final layer's output embedding to the prediction[1]. Trainable and frozen parameters are represented in different colors, respectively.

---

[1]The mapping of [CLS] token is a general procedure in ViT. However, there are some vision transformer architectures (*e.g.*, Swin [66], CaiT [101]) do not use [CLS] and utilize global pooled $Z_N$ as input for Head.

## 4 WHEN SHOULD WE CHOOSE VPT?

Our investigation starts with experimental analysis to identify the conditions that VPT are preferrable. We first introduce our experiment settings and then present our key findings.

### 4.1 EXPERIMENT SETUP

**Dataset** Our experiments are conducted on VTAB-1k [127] image classification benchmark, which encompasses 19 visual tasks categorized into three groups: *Natural*, *Specialized*, and *Structured* (*i.e.*, for image examples, refer to §D). The *Natural* group consists of seven datasets comprising natural images captured by standard cameras. The *Specialized* group includes four datasets that cover images taken by specialized equipment. The *Structured* group contains eight tasks that require geometric comprehensions, such as counting and distance measurement. Unless stated otherwise, we follow the standard experimental setup of [49; 127; 37] and use an 800-200 split for hyperparameter tuning, where 800 samples are used for training and 200 for validation. We evaluate the final performance on the full testing set. For the one-shot experiments, please refer to Appendix §G.4.
**Baselines** To ensure fair comparisons, we adopt the approach in VPT [49] and focus on the widely-used Vision Transformer (ViT) [24] for image classification. For the pre-training stage, we follow the default procedure outlined in [24; 86] and pre-train the vision Transformer on the ImageNet-21k dataset [89]. Our implementation and experiments are conducted using the publicly available model[2]. It should be noted that there are other backbone models (*e.g.*, Swin [66]) and training objectives (*e.g.*, MAE [41]) that can be used for this study. We have observed similar patterns.
**Implementation Details** Following [49; 37; 69; 15], we conduct grid search to match the best tuning hyperparameters, learning rate (*i.e.*, [50, 25, 10, 5, 2.5, 1, 0.5, 0.25, 0.1, 0.05]), and weight decay (*i.e.*, [0.01, 0.001, 0.0001, 0.0]) on `val` set for each task. Notably, we follow [37] and *do not cover* specific-designed large learning rate in [49] for general propose. In the training of all models, a cosine decay policy is employed to schedule the learning rate. The total number of training epochs is set to be 100 (including 10 warm-up epochs). We follow the same batch size setting: 64/128 for ViT-B/16. Our method is implemented in Pytorch [80]. Experiments are conducted on NVIDIA A100-80GB GPUs. For reproducibility, our full implementation will be publicly released.

### 4.2 INITIAL EXPERIMENTAL RESULTS

The initial results of FT and VPT are presented in the first two rows of Table 1 and Table 2 (*i.e.*, the last two rows: Mixed and FT-then-PT, will be introduced and compared to FT and VPT in §5.2). Out of all 19 datasets/tasks, VPT performs better on 16 instances, including 6/7 Natural datasets, 2/4 Specialized datasets, and 8/8 Structured datasets.

Table 1: **Image classification accuracy on various training strategies on VTAB-1k [127]** *Natural* **and** *Specialized* **for ViT-B/16 [24] pretrained on supervised ImageNet-21k [89]**. Underlines indicate the better results between FT and VPT, and **bolds** indicates the best results among all variants. We report five runs to test average accuracy instead of three in [49] to consider more randomness, and "Number of Wins" in (·) compared to full finetuning (Full) [47].

| ViT-B/16 [24] (85.8M) | VTAB-1k [127] *Natural* [7] | | | | | | | Mean | VTAB-1k [127] *Specialized* (4) | | | | Mean |
|---|---|---|---|---|---|---|---|---|---|---|---|---|---|
| | CIFAR-100 | Caltech101 | DTD | Flowers102 | Pets | SVHN | Sun397 | | Patch Camelyon | EuroSAT | Resisc45 | Retinopathy | |
| FT [47] | 64.5 | 88.1 | 65.0 | 97.3 | 84.6 | 87.3 | 39.5 | 75.19 | 81.2 | 95.7 | 83.2 | 73.3 | **83.48** |
| VPT [49] | 77.7 | **90.2** | **68.8** | **98.1** | **88.4** | 82.5 | **51.0** | **79.53**(6) | 77.9 | **96.2**(9) | **83.4** | 73.1 | 82.65(1) |
| Mixed | 61.5 | 86.2 | 60.2 | 95.6 | 85.0 | 85.7 | 35.4 | 72.80(1) | 77.8 | 96.2(7) | 74.0 | 71.0 | 79.78(1) |
| FT-then-PT | **85.9** | 89.0 | 66.3 | 95.9 | 86.3 | 87.1 | 36.9 | 78.20(4) | 79.4 | 95.6 | 81.8 | **74.0** | 82.70(1) |

Table 2: **Image classification accuracy on various training strategies on VTAB-1k [127]** *Structured* **for ViT-B/16 [24] pretrained on supervised ImageNet-21k [89]**.

| ViT-B/16 [24] (85.8M) | VTAB-1k [127] *Structured* [8] | | | | | | | | Mean |
|---|---|---|---|---|---|---|---|---|---|
| | Clevr/cnt. | Clevr/dist. | DMLab | KITTI/dist. | dSprites/loc. | dSprites/ori. | SmallNORB/az. | SmallNORB/elev. | |
| FT [47] | 55.4 | 58.2(2) | 40.4 | 74.7 | 54.0 | 47.0 | 26.2 | 28.6 | 48.07 |
| VPT [49] | **68.2** | 58.2(3) | **45.3** | **77.7** | **78.4** | 48.4 | **31.2** | **41.3** | **56.09**(8) |
| Mixed | 57.9 | 54.6 | 37.7 | 69.2 | 20.9 | 45.2 | 28.5 | 29.5 | 42.94(3) |
| FT-then-PT | 63.6 | **61.3** | 39.1 | 73.0 | 55.1 | **51.6** | 27.6 | 31.7 | 50.38(6) |

---

[2]https://github.com/google-research/vision_transformer

## 4.3 CHOOSE VPT FOR THREE QUADRANTS OF TRANSFER LEARNING

Upon initial examination, these results do not reveal a distinct pattern indicating when VPT outperforms FT. However, upon deeper reflection on the nature of transfer learning, the effectiveness of finetuning methods under the "pretrain-then-finetune" paradigm can be significantly impacted by the disparities between pretraining and finetuning. These disparities can be attributed to two key aspects: 1) the data distributions in the pretraining and finetuning datasets are dissimilar, and 2) the nature of the downstream task objective is different from the pretraining one.

Hence we analyze the datasets by categorizing them into four quadrants along these two dimensions. To measure the discrepancy in image distribution, we adopt the Fréchet Inception Distance (FID) [17; 56], which is known to correspond well with human judgments of image quality and diversity [116; 79]. In terms of task disparity, tasks in the *Natural* and *Specialized* subsets are closely related to image classification and thus have low disparities, while tasks in *Structured*, such as counting and distance, are regarded as distinct from image classification.

The FID scores with respect to the better finetuning method of all 19 tasks under the default train-val split are shown in Figure 3. The figure shows that VPT reaches superior performance than FT in two different scenarios: (1) when the disparity between the task objectives of the original and downstream tasks is high, or (2) when the data distributions are similar. Only for those tasks with low disparity (the left 11), a relatively larger FID score (dissimilar datasets) typically leads to higher performance of full finetuning compared to prompt tuning.[3]

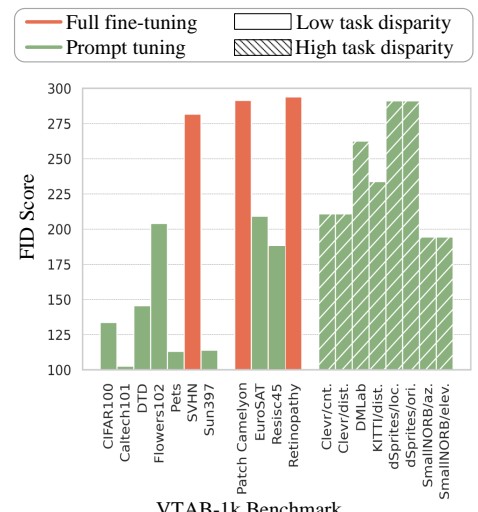

Figure 3: **Overall FID score with respect to win/loss of visual prompt tuning** on VTAB-1k benchmark categorized into *Natural*, *Specialized* and *Structured*, respectively. ▬▬ colors represent the method with higher accuracy. Under the same train-val split and low task disparity, a higher FID score might potentially lead to relatively higher accuracy on full finetuning, and vice versa. Solid filled represents that the disparity between the target task and the pretrained task is small while slash filled means a large disparity (*e.g.*, distance, azimuth, counting). The FID scores show significant robustness in repeat runs (*i.e.*, $std < 0.5\%$), we therefore do not present error bars here.

## 4.4 GAP NARROWS AS DOWNSTREAM DATASETS EXPAND

The above observation holds for scenarios when we have a limited amount of data for the downstream tasks. However, the effectiveness of different finetuning methods may vary in high-resource scenarios. To investigate this, we conducted a comprehensive evaluation of FT and VPT across different tuning set sizes. Specifically, we re-create the training and validation sets that keep a consistent 4:1 ratio, and vary the number of training samples from 400 to 20,000 to explore the behavior of the FT and VPT when the tuning data size grows. It is important to note that not all datasets support all combinations of splits due to insufficient data samples (*e.g.*, VTAB-1k *Natural* Oxford Flowers102 [73] has only 1020 images for both `train` and `val`).

The performance of full finetuning and prompt tuning across different training set sizes and datasets is presented in Figure 4. The figure shows that the performance gap between full finetuning and prompt tuning decreases as the amount of training data increases, with full finetuning even outperforming prompt tuning under high-resource settings. These results demonstrate the robustness and generality of prompt tuning in the face of limited finetuning examples. Although full finetuning generally achieves higher accuracy when rich data examples are available, the prompt tuning still

---

[3]Here the range of FID scores reported appears to be larger than existing approaches [99; 17; 79; 128]. This is because current methods typically report the distance between real and generated images, whereas our analysis involves the computation of distances between real datasets with significantly different distributions.

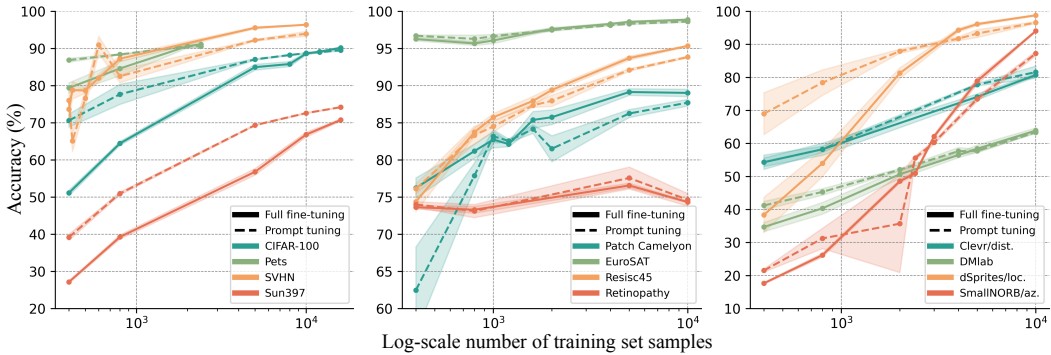

Figure 4: **Analysis of dataset capacity** on VTAB-1k *Natural* (left), *Specialized* (middle) and *Structured* (right), respectively. For each group, we select four datasets and plot accuracy plots on FT (*i.e.*, solid lines) and VPT (*i.e.*, dotted lines). We take the log scale of training data samples for better separation and each color stands for an individual classification task. Each point is given by average over five runs. In general, with the dataset increasing in size, the performance gap between FT and VPT becomes narrow. FT even surpasses VPT in 9 of 12 cases in this plot with the increasing of data samples (the same tendency takes place in other datasets). For per-task accuracy tables among different dataset scales and detailed FT and VPT accuracy plots, see the supplementary material §A.

Table 3: **One-shot image classification accuracy on VTAB-1k [127]** *Natural* and *Specialized* for ViT-B/16 [24] pretrained on supervised ImageNet-21k [89]. Experiments show that VPT outperforms the FT in **13 of 19** instances under the one-shot learning setting, which further suggests the advantage of VPT in limited data cases. "Tuned/Total" is the average percentage of tuned parameters required by 19 tasks.

| ViT-B/16 [24] (85.8M) | VTAB-1k [127] *Natural* [7] | | | | | | | Mean | VTAB-1k [127] *Specialized* [4] | | | | Mean |
|---|---|---|---|---|---|---|---|---|---|---|---|---|---|
| | CIFAR-100 | Caltech101 | DTD | Flowers102 | Pets | SVHN | Sun397 | | Patch Camelyon | EuroSAT | Resisc45 | Retinopathy | |
| FT [47] | 37.4 | 67.9 | 25.3 | 80.2 | 37.8 | **12.7** | 30.3 | 41.66 | 58.2 | 44.1 | **42.3** | 32.6 | 44.30 |
| VPT [49] | **54.6** | **78.4** | **37.4** | **94.9** | **70.5** | 11.6 | **51.2** | **56.94**(6) | **68.4** | **45.6** | 41.1 | **34.7** | **47.45**(3) |
| - Tuned / Total | 0.20% | 0.20% | 0.15% | 0.10% | 0.04% | 0.54% | 0.41% | 0.23% | 1.06% | 1.07% | 0.15% | 0.02% | 0.57% |

reaches a competitive performance with much fewer parameters. This observation aligns with current research in NLP prompt tuning [13; 22] and suggests that prompt tuning techniques have great potential in both low- and high-resource scenarios.

To further test the effectiveness of the two methods in low-resource settings, we conducted one-shot learning experiments where each class is learned from a single labeled example [104; 98]. These experiments represent an extreme case in which we seek to evaluate the methods' ability to adapt to novel tasks after having been exposed to only a small amount of training data. Table 3 and Table 4 show the experimental results, with prompt tuning outperforming full finetuning in **13 out of 19** tasks and achieving substantially higher accuracy in some cases (*i.e.*, 41.66% *vs* **56.94%** in VTAB-1k *Natural*). These results favor our assumption that prompt tuning is more effective than full finetuning for the aforementioned three quadrants when only limited finetuning examples are available. We observe that in only half of the Structured cases, VPT performs better. Overall, the accuracies for one-shot experiments are comparatively low. Hence it should be noted that randomness could play a role in one-shot experiments, and the results might not reliably conclusive.

## 5   WHY DOES VPT OUTPERFORM FT?

### 5.1   OVERFITTING IS PART OF THE REASON

After identifying the conditions that VPT would be favorable over FT, we dive deeper to explore the underlying reasons. One hypothesis is that FT optimizes all the parameters in the backbone, which may easily overfit to the downstream task, especially when limited data is available [121; 21; 39].

Figure 5 presents training and testing loss curves, up to 100 epochs, for six representative tasks in VTAB-1k [127], comprising of 2 tasks for from each kind (*Natural*, *Specialized*, and *Structured*). Comprehensive results covering all 19 tasks are available in the supplementary material §A.

We observe that for 8 out of 9 *Structured* tasks, both FT and VPT overfit (*i.e.*, testing error increases as training progresses). These tasks, such as counting and angular measurement, are very different

Table 4: **One-shot image classification accuracy on VTAB-1k [127]** *Structured* for ViT-B/16 [24] pretrained on supervised ImageNet-21k [89].

| ViT-B/16 [24]
(85.8M) | VTAB-1k [127] *Structured* [8] | | | | | | | | Mean |
|---|---|---|---|---|---|---|---|---|---|
| | Clevr/cnt. | Clevr/dist. | DMLab | KITTI/dist. | dSprites/loc. | dSprites/ori. | SmallNORB/az. | SmallNORB/elev. | |
| FT [47] | 17.8 | 21.0 | 17.1 | **40.5** | **9.0** | **11.4** | **7.4** | 12.7 | **17.11** |
| VPT [49] | **21.5** | **22.1** | **18.7** | 37.0 | 6.7 | 7.5 | 6.2 | **14.9** | 16.83(4) |
| - Tuned / Total | 0.54% | 2.11% | 1.07% | 0.54% | 0.12% | 0.55% | 2.12% | 2.11% | 1.14% |

Figure 5: **Training/testing loss curves** of six datasets from VTAB-1k. █ █ █ █ colors represent four training strategies: full finetuning, prompt tuning, mixed, and FT-then-PT, respectively. We show two representative tasks from VTAB-1k *Natural*, *Specialized*, and *Structured*, respectively. Full results and log scale results are presented in the supplementary material §A.

from image classification, and may require more training data for model adaptation. We conducted additional experiments by enlarging the training set (to 5000 and 10000) of these tasks in Figure 6. It can be observed that with the increase of training samples, the phenomenon of overfitting is mitigated in both methods.

However, **among the remaining 10 tasks within the categories of** *Natural* **and** *Specialized***, overfitting behavior is observed in only 1 task**. In other words, training and testing losses consistently decrease for both methods in scenarios where the task objectives are similar between the original and downstream tasks. In such cases, overfitting is not the cause of performance degradation of FT.

## 5.2 OPTIMIZATION DOES NOT PARTICULARLY FAVOR VPT

Another possible explanation for why prompt tuning can achieve superior performance is that it is the additional dimensions introduced by the prompts that help the loss to be further optimized during the tuning process. This hypothesis is illustrated by Figure 7.

To test our hypothesis in depth, we perform experiments on two additional variants of tuning strategies besides full finetuning and prompt tuning. The first method is noted *Mixed*, where we add visual prompts and perform full finetuning on the entire network instead of focusing only on network parameters or prompt vectors in isolation. The second method is called *FT-then-PT*, where

we first perform full finetuning on the network for the downstream task and then prepend additional prompts and perform prompt tuning for the same task. This approach applies 200 epochs in total.

We compare the above methods with full fine-tuning and visual prompt tuning in Figure 5. Per-task performance is also detailed in Table 1 and Table 2. We observe that in general, *VPT* demonstrates the highest performance. The *FT* approach and the *FT-then-PT* approach yield similar results, albeit with a noticeable gap compared to *VPT*. In contrast, the *Mixed* approach exhibits the lowest performance among the evaluated methods.

These results indicate several key findings. (1) The presence of additional dimensions **does not** significantly aid the optimization process in escaping local minima. If this were the case, we would expect either the *Mixed* or the *FT-then-PT* method to outperform *FT*, but this is not observed. (2) Preserving the original feature proves to be crucial for transferring to downstream tasks, as evidenced by the superior performance of *VPT* compared to *FT-then-PT*. (3) Maintaining a fixed feature space in *VPT* may compel it to learn more valuable embeddings with the trainable prompts, as it significantly outperforms the *Mixed* method. These observations lead to an important insight: pretrained parameters play a pivotal role in capturing general features, while the added parameters are important for potentially encoding task information in the context of transfer learning.

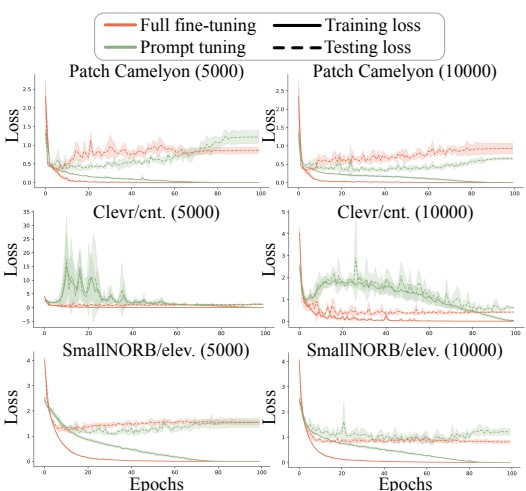

Training/testing loss curve for VTAB-1k overfitting datasets

Figure 6: **Training/testing loss curves for VTAB-1k *Structured* datasets with larger training sets.** colors represent full finetuning and prompt tuning, respectively. Solid lines are the training loss curves and dashed lines show the corresponding testing curves. We select three tasks that suffer from overfitting and enlarge the dataset train-val split to 5000-1250 (left) and 10000-2500 (right), respectively. Full results are shown in the supplementary material §A.

## 5.3 FURTHER OBSERVATIONS

Overall, our observations indicate that the preservation of initial features is important for VPT's success, but in a very sophisticated manner. Notably, VPT surpasses more straightforward methods of feature preservation, such as retraining the final linear probe layer or incorporating a multi-layer probe [37], even though all these methods maintain the originally learned features.

This potentially explains cases in which data distributions exhibit similarity, but not in scenarios characterized by distinct data distributions and high task disparity. In these instances, we hypothesize that the substantial task divergence requires a stronger need for additional finetuning examples to achieve comprehensive adaptation, whereas prompt tuning demonstrates better generalization. The introduced prompts serve as guiding mechanisms for the model to enhance task-specific feature learning, a pivotal aspect of task encoding, particularly in situations with limited finetuning data.

It's also worth discussing scenarios in which FT outperforms VPT, specifically in situations involving similar tasks with varying data distributions. Our hypothesis is that the feature representation captured by the pretrained model may not be well-suited for the downstream task due to significant differences in data distribution. Consequently, updating all model parameters through full finetuning becomes more effective for learning the feature representation within

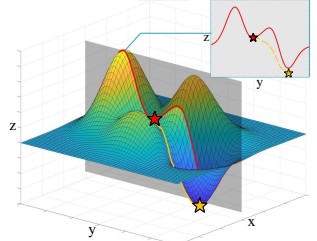

Figure 7: Hypothesis of extra dimensions helping the pretrained model to be further optimized, *e.g.*, by escaping local minima. In this illustration, pre-training the original 1D function (the red line) might get stuck in the middle (the red star). When a second dimension is introduced during finetuning, it is easier to be better optimized (from the yellow line to the yellow star). However, we found that this hypothesis does not hold.

the new distribution. Conversely, when finetuning data is similar, there may be no need to modify the backbone model extensively; instead, a minimal set of prompts can be inserted for adaptation.

## 5.4 VISUALIZING THE EFFECT OF VPT ON FEATURE LEARNING

To the best of our knowledge, the visual explanations of visual prompt tuning are found to be rare [60; 75; 71]. In light of this view, we seek to investigate whether prompt tuning can offer actual visualization meanings and support stronger visualization explanations, particularly in cases where it outperforms full finetuning. In this experiment, we apply gradient-weighted class activation mapping (GradCAM[4]) [92], which is a popular technique for producing visual explanations for decisions [61; 109; 2; 10; 35; 29; 11]. In general, it utilizes the gradients of a target concept and channels them through the final layer of the network to generate a coarse localization map that accentuates the salient regions in the image which contribute significantly towards predicting the concept of interest.

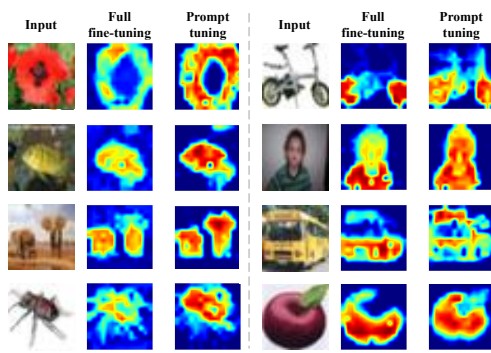

Figure 8 presents examples that full finetuning fails to recognize while prompt tuning makes correct classification during testing. It can be seen from the results that a clear concept of interest can be observed from both methods through a heat map. In these prompt tuning success cases, we can see straightforward visual explanation differences (*e.g.*, take the bicycle image in Figure 8 upper right as an example, when full finetuning fails to make a correct decision, prompt tuning instead recognize the bicycle with its structural features successfully), which suggests that prompt tuning effectively guides the model to pay more attention to the important regions of the images, and thus is capable of enhancing the learning of stronger features by leveraging domain-specific information and patterns that might not be well captured in full finetuning. An additional visual explanation technique (*i.e.*, Integrated Gradients [100]) and more visual inspection examples on GradCAM are also included in the supplementary material §B.

Figure 8: **Visual inspection of full finetuning and prompt tuning** using GradCAM [92]. Note that red regions correspond to a high score for the class. From left to right are input image after standard data augmentation, GradCAM results for full finetuning and GradCAM results for prompt tuning. Figure best viewed in color.

## 6 CONCLUSION AND DISCUSSION

When models scale up, how shall we face the elephant in the room? Driven by this question, we aim to provide a justifiable recommendation regarding the choice between full finetuning or prompt tuning during training. In this pursuit, we conduct an extensive study across 19 datasets, comparing full finetuning with visual prompt tuning, to examine various hypotheses regarding the effectiveness of visual prompt tuning. Our experimental results *show* that overfitting is not the root cause of full finetuning's inferior performance to prompt tuning. We further notice that dataset disparities might be a potential factor in the behavior of full finetuning and prompt tuning. Also, prompt tuning shows its strong generality with limited data, while full finetuning catches up or surpasses prompt tuning with more data presented. Attention map visualization suggests that the visual prompts are capable of enhancing the learning of features that might not be well captured in full finetuning. We *suggest* that prompt tuning should be applied consciously under "pretrain-then-finetune" paradigm. Task disparities and dataset scale are two main factors that determine the most suitable way for downstream finetuning. In spite of our comprehensive coverage of various datasets in image recognition, it is noteworthy that other visual-related tasks for finetuning (*e.g.*, semantic segmentation [62; 28]) have not been fully explored. This observation motivates us for future investigation of such scenarios.

---

[4]https://github.com/pytorch/captum

## REPRODUCIBILITY STATEMENT

To help readers reproduce our results, we have described the implementation details in §4.1 and §G. We will release our source code after acceptance. All the datasets we use are publicly available.

## ACKNOWLEDGMENT

This research was supported by the National Science Foundation under Grant No. 2242243.

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

## SUMMARY OF THE APPENDIX

This supplementary contains additional details for the twelfth International Conference on Learning Representations submission, titled *"Facing the Elephant in the Room: Prompt-Tuning or Full finetuning?"*. The supplementary is organized as follows:

- §A shows **comprehensive training/testing curve** for VTAB-1k benchmark and cover comprehensive experiments on overfitting datasets.

- §B presents **visualization inspections** on a new-added explanation method — Integrated Gradients, and more results on GradCAM.

- §C shows the per-task results of full finetuning and visual prompt tuning on different **backbone and pretraining objectives**.

- §D presents the **image examples from the VTAB-1k image classification benchmark**.

- §E shows the optimal **prompt length** for each task in VTAB-1k image classification benchmark.

- §F further presents the **FID scores from a new-added image classification benchmark — FGVC [37], and compare the performance between visual prompt tuning and full fine-tuning.**

- §G is the **discussion** of legal/ethical considerations, reproducibility, social impact, limitations and future work.

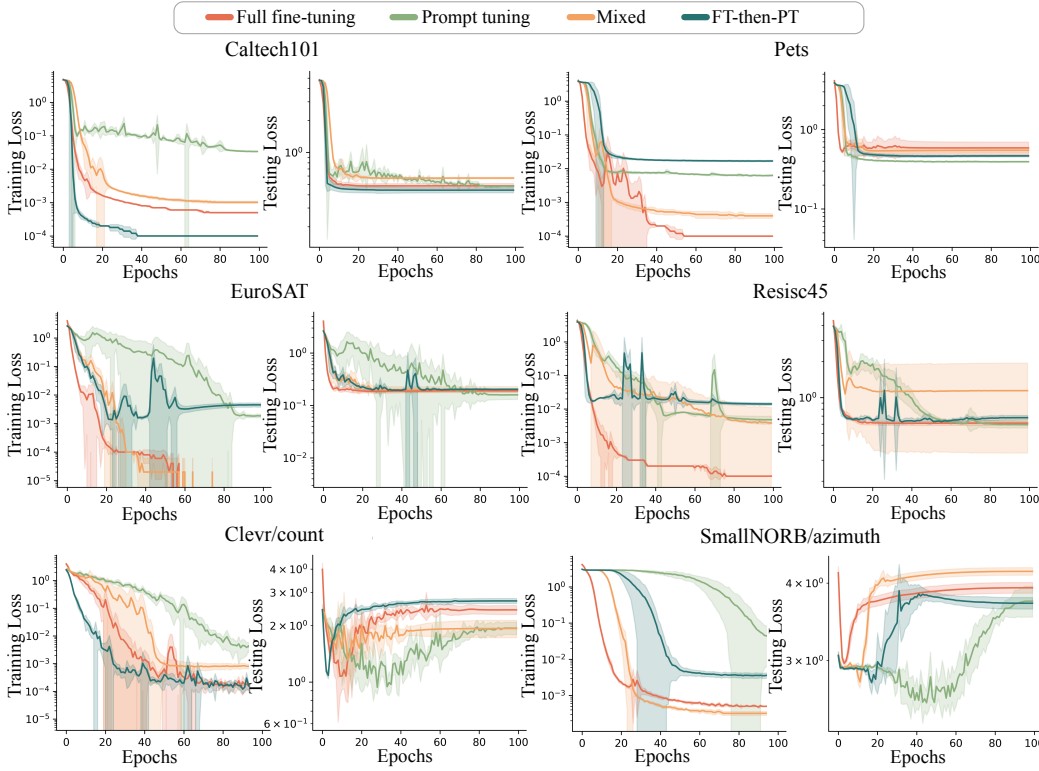

Figure 9: **Log-scale version of Figure 5**. Consistent to our paper, ▬▬▬▬ colors represent four training strategies: full finetuning, prompt tuning, mixed and FT-then-PT, respectively. Same for Figure 10, Figure 11 and 12.

## A    PER-TASK TRAINING/TESTING CURVE

In Figure 9, we further provide the log-scale version of Figure 5.

In Figure 10, 11 and 12, we comprehensively present per-task training/testing curve on VTAB-1k [127] *Natural*, *Specialized* and *Structured*, respectively. We do not observe overfitting for full

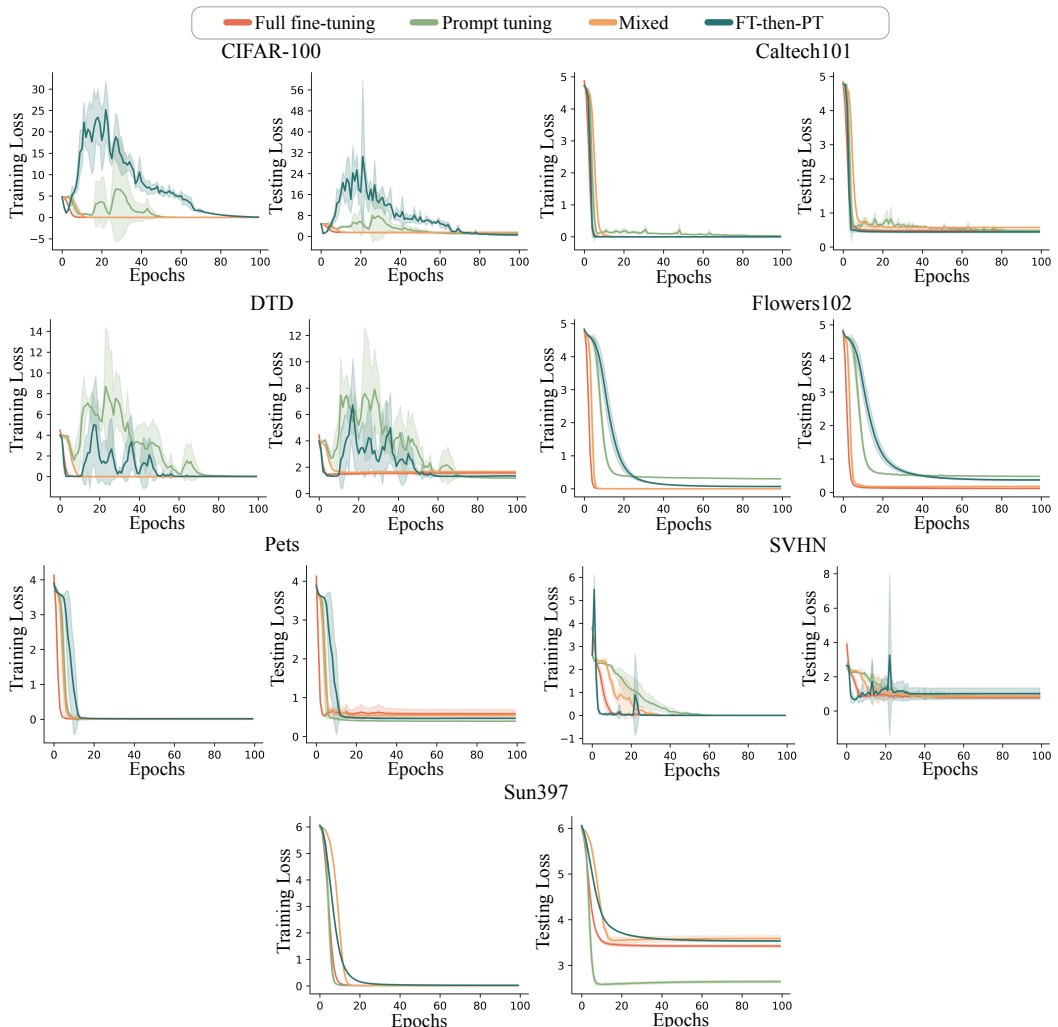

Figure 10: **Per-task training/testing loss curve for VTAB-1k [127]** *Natural*.

finetuning in 10 of 19 cases, while 9 of 19 datasets suffer from overfitting for both full finetuning and prompt tuning. Further experiments in Figure 13 on increasing training samples reduce overfitting for both methods. Overall, it can be inferred that overfitting is not the underlying cause of performance degradation observed in full finetuning when compared to prompt tuning.

In Table 5 and 6, we provide per-task results in accuracy with different dataset scales (See Figure 2, 4 and §5.1 in our paper. Not all datasets are provided in the same scale due to the limit number of data samples). We further provide comprehensive accuracy curves with different dataset scales in Figure 14, 15 and 16 for VTAB-1k *Natural*, *Specialized* and *Structured*, respectively, which covers more combinations that are not showed in Table 5 and 6. The figures and Tables are consistent with our observations and support our assumption in the main paper that the performance gap between full finetuning and prompt tuning decreases as the amount of training data increases, with full finetuning even outperforming prompt tuning under high-resource settings, demonstrating the robustness and generality of prompt tuning in the face of limited finetuning examples. Though full finetuning generally achieves higher accuracy when rich data examples are available, the prompt tuning still has a competitive performance with much fewer parameters.

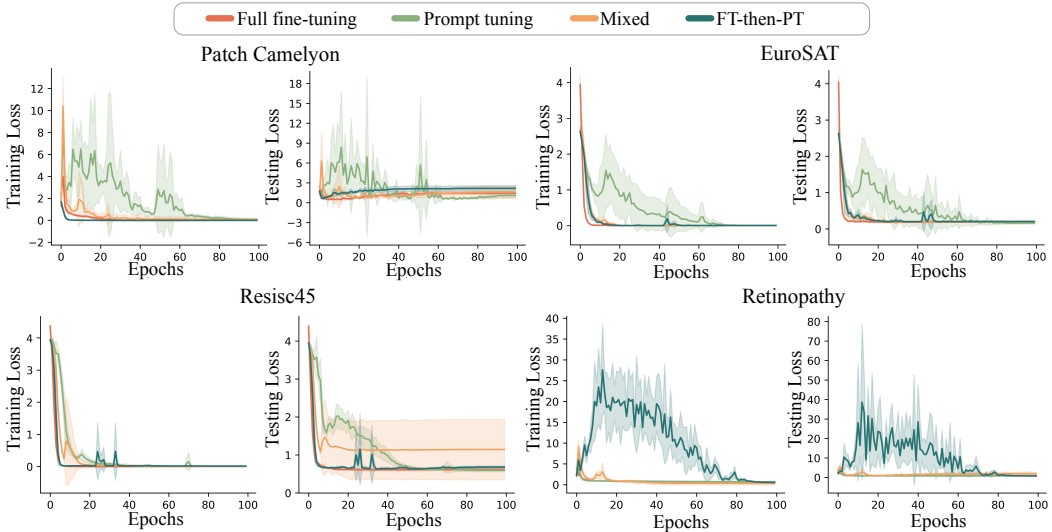

Figure 11: **Per-task training/testing loss curve for VTAB-1k [127]** *Specialized*.

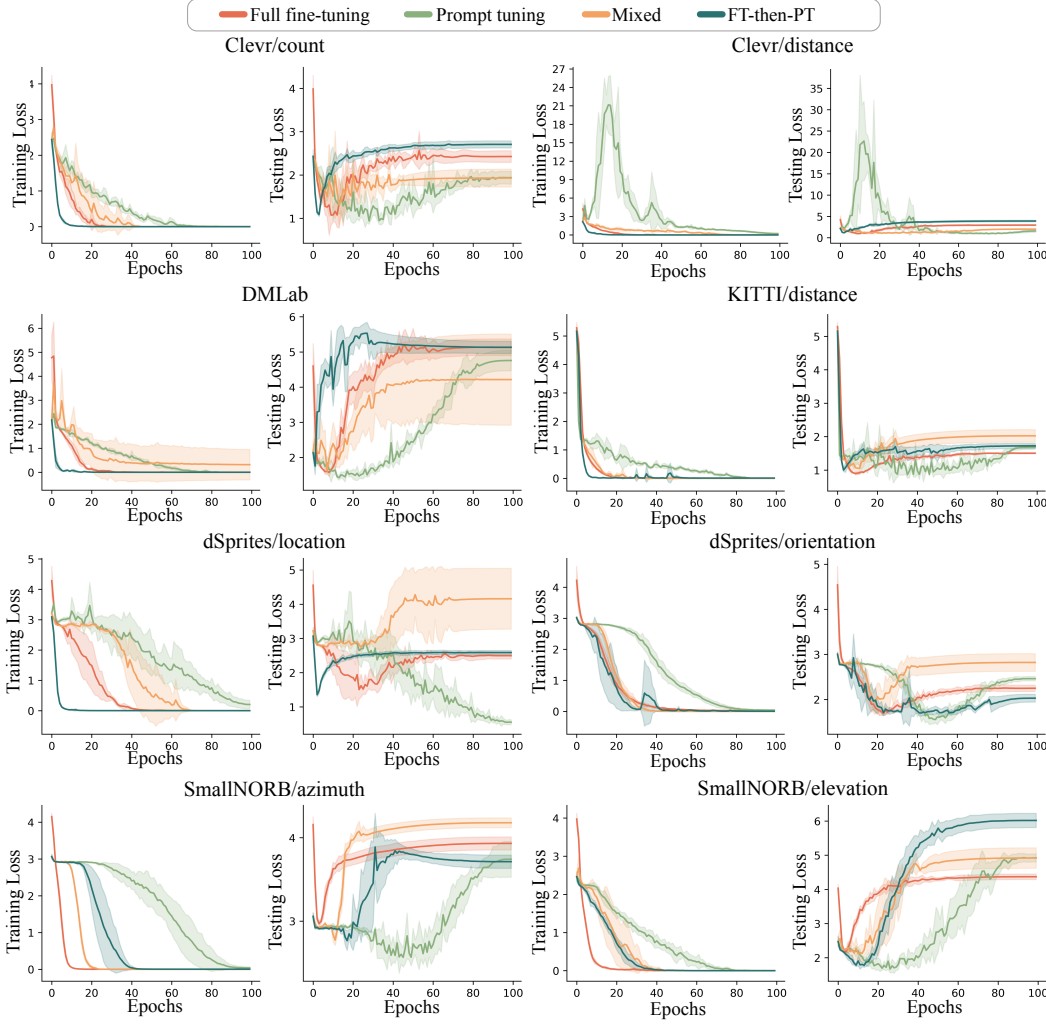

Figure 12: **Per-task training/testing loss curve for VTAB-1k [127]** *Structured*.

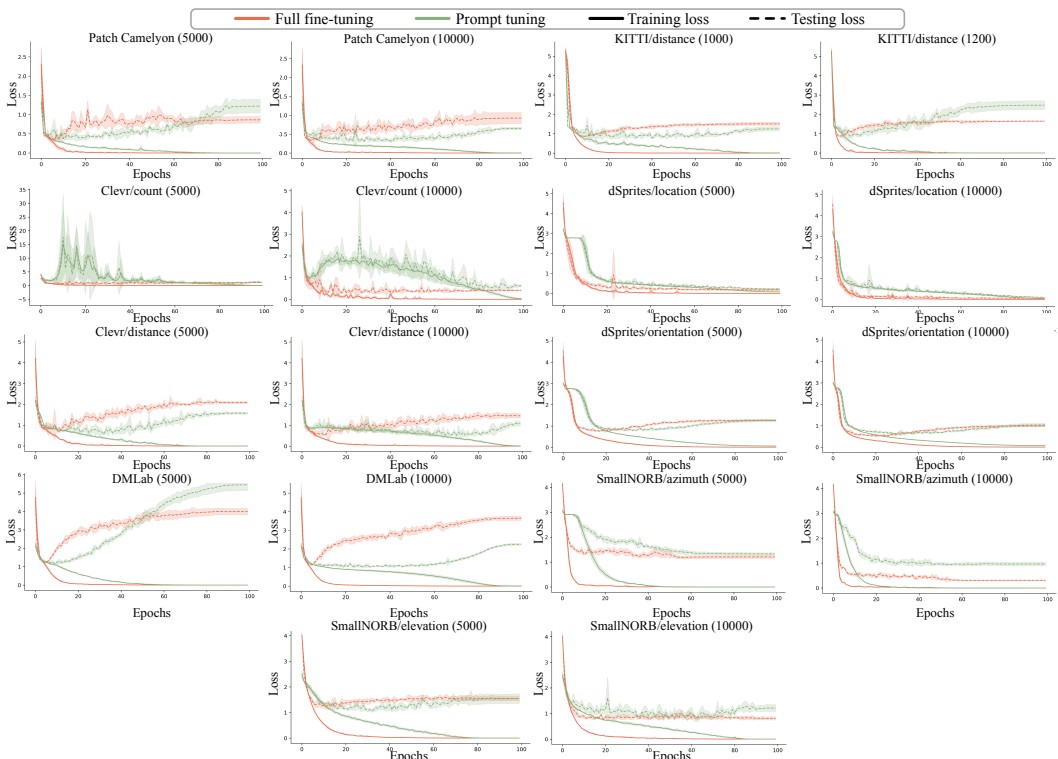

Figure 13: **Training/testing loss curve for VTAB-1k [127] when increasing training samples**. Note that the listed 9 cases are observed in overfitting on both full finetuning and prompt tuning under default numbers of training samples. We further increase the number of training samples and find that both methods are mitigated from overfitting. Further enlarging in training data results in unacceptable training time and fails to cover some cases in this plot (*i.e.*, not having enough training samples). (·) represents the number of training samples. For most cases, we apply 5000/10000 number of images for training. VTAB-1k [127] *Structured* KITTI/distance [32], on the other hand, does not have sufficient data for training, we present results for 1000/1200.

Table 5: **Image classification accuracy on different scales on VTAB-1k [127] *Natural* and *Specialized*** for ViT-B/16 [24] pretrained on supervised ImageNet-21k [89]. Note that not all combinations in Figure 4 are listed in this table. "# Samples" represent the number of training samples during finetuning. We pick 400, 800, 5000 and 10000 number of training data as the general number of samples across all datasets. All results are averaged on five runs. Same for Table 6.

| ViT-B/16 [24] (85.8M) | # Samples | VTAB-1k [127] *Natural* [7] | | | | | | | Mean | VTAB-1k [127] *Specialized* (4) | | | | Mean |
| --- | --- | --- | --- | --- | --- | --- | --- | --- | --- | --- | --- | --- | --- | --- |
| | | CIFAR-100 | Caltech101 | DTD | Flowers102 | Pets | SVHN | Sun397 | | Patch Camelyon | EuroSAT | Resisc45 | Retinopathy | |
| FT [47] | 400 | 51.1 | 76.1 | 55.1 | 92.9 | 79.4 | 73.6 | 27.1 | 65.04 | 76.2 | 96.3 | 74.4 | 73.6 | 80.13 |
| VPT [49] | 400 | 70.6 | 84.9 | 60.6 | 97.5 | 86.9 | 76.0 | 39.2 | 73.67 | 62.5 | 96.7 | 76.1 | 74.0 | 77.33 |
| FT [47] | 800 | 64.5 | 88.1 | 65.0 | 97.3 | 84.6 | 87.3 | 39.5 | 75.19 | 81.2 | 95.7 | 83.2 | 73.3 | 83.48 |
| VPT [49] | 800 | 77.7 | 90.2 | 68.8 | 98.1 | 88.4 | 82.5 | 51.0 | 79.53 | 77.9 | 96.2(9) | 83.3 | 73.1 | 82.65 |
| FT [47] | 5000 | 85.0 | - | - | - | - | 95.6 | 56.8 | 79.13 | 89.1 | 98.6 | 93.7 | 76.6 | 89.50 |
| VPT [49] | 5000 | 87.1 | - | - | - | - | 92.2 | 69.4 | 82.9 | 86.3 | 98.3 | 92.1 | 74.3 | 87.75 |
| FT [47] | 10000 | 88.6 | - | - | - | - | 96.4 | 66.8 | 83.93 | 89.0 | 98.9 | 95.3 | 77.6 | 90.20 |
| VPT [49] | 10000 | 88.7 | - | - | - | - | 94.0 | 72.6 | 85.10 | 87.7 | 98.6 | 93.9 | 74.6 | 88.70 |

## B VISUALIZATION INSPECTIONS

We further introduce Integrated Gradients [100] (IG), and present visualization inspections in Figure 17(a) to support the advantages of visual prompts discussed in our paper. In general, IG tries to attribute the prediction of a deep network to its input features, and it is commonly applied [61; 2; 4] in the research of explainable Artificial Intelligence (AI). IG gains widespread adoption as an interpretability technique, owing to its versatility in explaining any differentiable model, including

Table 6: **Image classification accuracy on different scales on VTAB-1k [127]** *Structured* for ViT-B/16 [24] pretrained on supervised ImageNet-21k [89].

| ViT-B/16 [24] (85.8M) | # Samples | VTAB-1k [127] *Structured* [8] | | | | | | | | Mean |
| --- | --- | --- | --- | --- | --- | --- | --- | --- | --- | --- |
| | | Clevr/cnt. | Clevr/dist. | DMLab | KITTI/dist. | dSprites/loc. | dSprites/ori. | SmallNORB/az. | SmallNORB/elev. | |
| FT [47] | 400 | 45.3 | 54.3(1) | 34.7 | 72.2 | 38.4 | 32.6 | 17.7 | 23.3 | 39.82 |
| VPT [49] | 400 | 60.8 | 54.3(1) | 41.2 | 67.0 | 69.0 | 38.4 | 21.5 | 29.4 | 47.70 |
| FT [47] | 800 | 55.4 | 58.2(2) | 40.4 | 74.7 | 54.0 | 47.0 | 26.2 | 28.6 | 48.07 |
| VPT [49] | 800 | 68.2 | 58.2(3) | 45.3 | 77.7 | 78.4 | 48.4 | 31.2 | 41.3 | 56.09 |
| FT [47] | 5000 | 84.4 | 74.1 | 58.4 | - | 96.1 | 73.1 | 78.9 | 67.95 | 66.62 |
| VPT [49] | 5000 | 76.2 | 77.8 | 57.8 | - | 93.3 | 70.2 | 73.5 | 68.02 | 73.83 |
| FT [47] | 10000 | 93.4 | 80.7 | 63.9 | - | 98.8 | 79.4 | 94.0 | 83.4 | 84.80 |
| VPT [49] | 10000 | 84.4 | 81.6 | 63.4 | - | 96.6 | 76.7 | 87.2 | 80.3 | 81.46 |

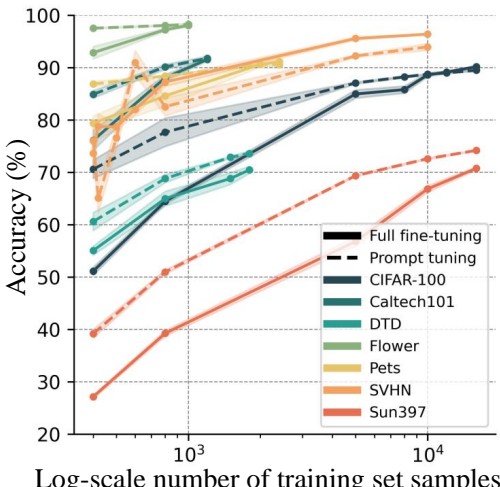

Figure 14: **Analysis of dataset capacity on VTAB-1k [127]** *Natural.* The solid lines stand for full finetuning and the dotted lines represent prompt tuning. We take the log-scale of training data samples for better separation and each color stands for an individual classification task. Each point is given by average over five runs. In general, with the increasing of dataset in size, the performance gap between full finetuning and prompt tuning becomes closer. Same for Figure 15 and 16.

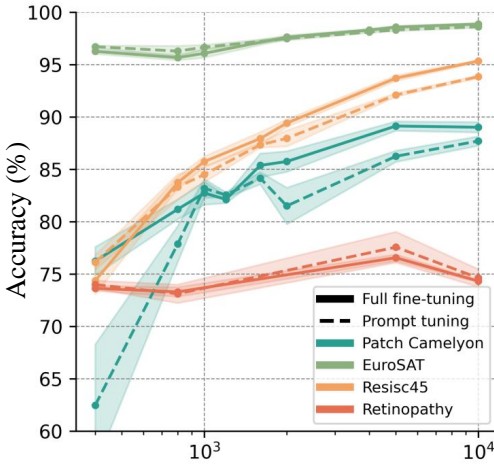

Figure 15: **Analysis of dataset capacity on VTAB-1k [127]** *Specialized.*

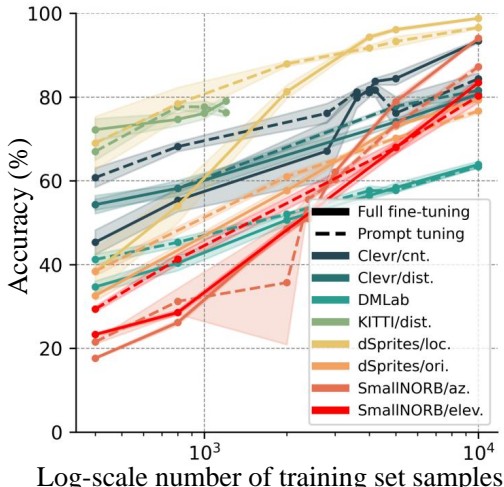

Figure 16: **Analysis of dataset capacity on VTAB-1k [127]** *Structured*.

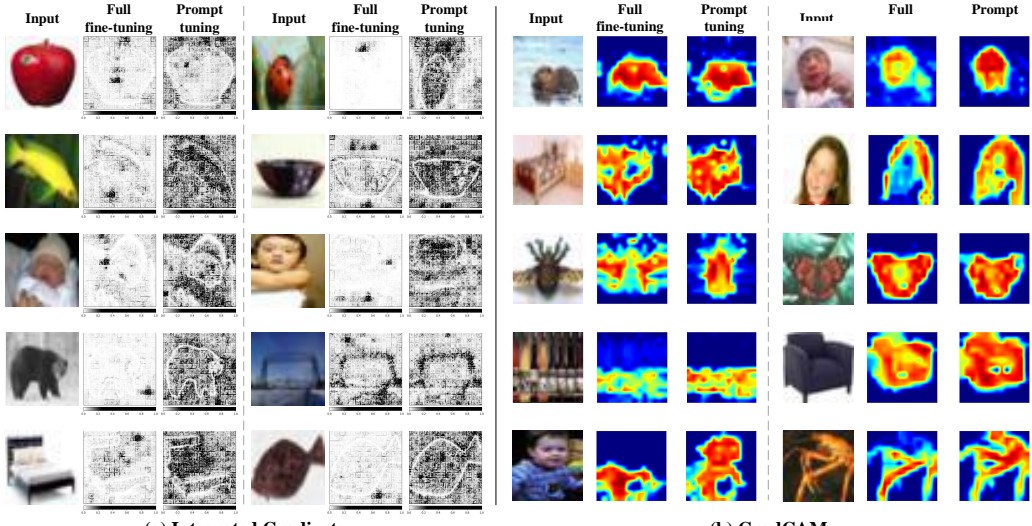

(a) Integrated Gradients       (b) GradCAM

Figure 17: **(a) Integrated Gradients (IG) [100] visual inspection of full finetuning and prompt tuning**. Note that the darker regions responds to high score for class (In light of the observation that certain images exhibit a consistently negative gradient, it becomes necessary to take the absolute value in order to ensure consistency and save successfully across all images within the testing set. This approach results in the emergence of darker boundaries within the resulting images). From left to right are input images after standard data augmentation, IG results for full finetuning and IG results for prompt tuning. **(b) More visual inspection of full finetuning and prompt tuning** using GradCAM [92]. Consistent to our paper, the red regions correspond to high score for class. From left to right are input image after standard data augmentation, GradCAM results for full finetuning and GradCAM results for prompt tuning. Figure best viewed in color.

images [100; 4], text [18; 84], and structured data [72; 122; 1]. In Figure 17(a), we can observe straightforward visual explanation differences (*e.g.*, take the Coccinella septempunctata image as an example, when full finetuning fails to provide high gradients to make a correct decision, prompt tuning instead recognize it with significantly higher gradients), showing consistency with our paper.

In Figure 17(b), we present more visualization inspection results for full finetuning and prompt tuning using GradCAM [92]. Overall, we present additional visual evidence to support the notion that prompt tuning encompasses actual visual explanations throughout the training process.

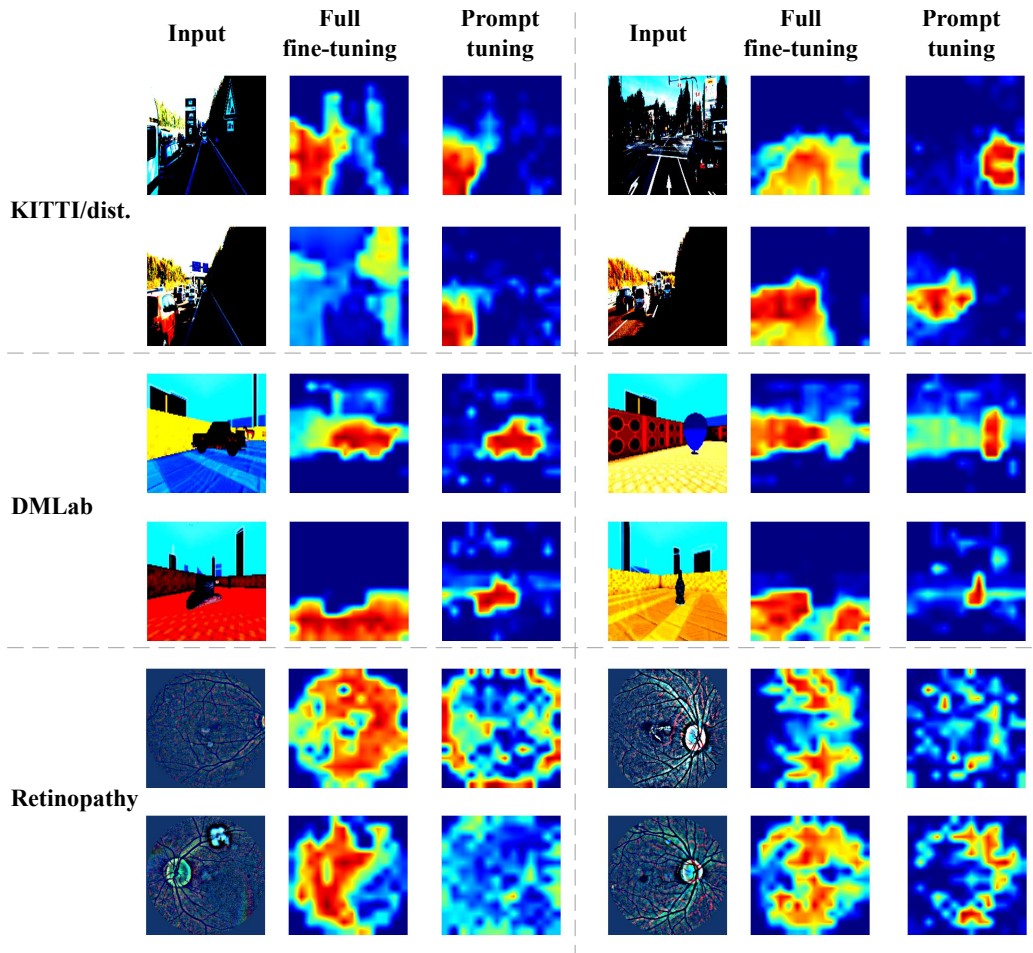

Figure 18: **Visual inspection of full finetuning and prompt tuning in other 3 groups.** For KITTI-dist. and DMLab, we present the cases where full finetuning gets inferior performance to visual prompt tuning and vice versa for Retinopathy. KITTI-dist. lies in the 2nd quadrant, DMLab lies in the 3rd quadrant and Retinopathy lies in the 4th quadrant with regard to Figure 1. Figure best viewed in color.

Table 7: **VTAB-1k [127] *Natural* per-task results for Swin-Base [66]** pretrained on supervised ImageNet-21k. Best results among full finetuning and prompt tuning are **bold**. We report the "number of wins" in [·] compared to full finetuning. Same for Table 8 to 15.

| Swin-Base [66] (86.7M) | VTAB-1k [127] *Natural* (7) | | | | | | | Mean |
|---|---|---|---|---|---|---|---|---|
| | CIFAR-100 | Caltech101 | DTD | Flowers102 | Pets | SVHN | Sun397 | |
| FT [47] | 72.2 | 88.0 | 71.2 | 98.3 | 89.5 | **89.4** | 45.0 | 79.10 |
| VPT [49] | **79.6** | **90.8** | **78.0** | **99.5** | **91.4(3)** | 86.4 | **51.7** | 78.78 (6) |
| - Tuned / Total (%) | 0.13 | 0.13 | 0.07 | 0.13 | 0.06 | 0.70 | 0.48 | 0.28 |

We further present visualization inspection results based on Figure 1. In Figure 8 and 17, we primarily focus on images with similar data distribution and task, we thus further posit visualization inspection results under the setting of other three groups in Figure 18. As seen, for KITTI-dist. and DMLab from 2nd and 3rd quadrant respectively, visual prompt tuning presents clear dense visual evidence and focuses on correct positions. On the other hand, for Retinopathy lies in the 4th quadrant, full finetuning presents more stable results, focusing on cells' characteristics.

Table 8: **VTAB-1k [127]** *Specialized* **per-task results for Swin-Base [66]** pretrained on supervised ImageNet-21k.

| Swin-Base [66] (86.7M) | VTAB-1k [127] *Specialized* [4] | | | | Mean |
|---|---|---|---|---|---|
| | Patch Camelyon | EuroSAT | Resisc45 | Retinopathy | |
| FT [47] | **86.6** | **96.9** | **87.7** | **73.6** | 86.21 |
| VPT [49] | 80.1 | 96.2 | 85.0 | 72.0 | 83.33 (0) |
| - Tuned / Total (%) | 0.07 | 0.13 | 0.19 | 0.02 | 0.10 |

Table 9: **VTAB-1k [127]** *Structured* **per-task results for Swin-Base [66]** pretrained on supervised ImageNet-21k.

| Swin-Base [66] (86.7M) | VTAB-1k [127] *Structured* [8] | | | | | | | | Mean |
|---|---|---|---|---|---|---|---|---|---|
| | Clevr/cnt. | Clevr/dist. | DMLab | KITTI/dist. | dSprites/loc. | dSprites/ori. | SmallNORB/az. | SmallNORB/elev. | |
| FT [47] | **75.7** | **59.8** | **54.6** | **78.6** | **79.4** | **53.6** | **34.6** | **40.9** | 59.65 |
| VPT [49] | 67.6 | 59.4 | 50.1 | 61.3 | 74.4 | 50.6 | 25.7 | 25.7 | 51.85 (0) |
| - Tuned / Total (%) | 0.70 | 0.70 | 0.14 | 0.69 | 0.15 | 0.09 | 0.16 | 0.02 | 0.38 |

## C  PER-TASK RESULTS ON DIFFERENT PRETRAINING OBJECTIVES.

We also report the per-task results on VTAB-1k [127] Swin-Base [66] (*i.e.*, Table 7, 8 and 9), MAE [41] (*i.e.*, Table 10, 11 and 12) and MoCo v3 [16] (*i.e.*, Table 13, 14 and 15), respectively. Overall, the empirical results consistently reveal the attainment of superior or competitive performance gains through prompt tuning, in comparison to full finetuning, across diverse tasks within the default train-val split. Notably, these performance gains are achieved while maintaining a substantially reduced number of model parameters.

## D  IMAGE EXAMPLES

In Figure 19, we include image examples from VTAB-1k [127] image classification benchmark, including 19 visual tasks categorized into three groups: *Natural*, *Specialized*, and *Structured*.

## E  PROMPT LENGTH

In Table 16, 17 and 18, we posit the per-task prompt length, which is consistent with the original VPT [37] approach.

## F  FID ON FGVC

In Table 19, we further present the corresponding FID scores on FGVC [37] image classification benchmark. Specifically, it contains 5 Fine-Grained Visual Classification, including CUB-200-2011 [106], NABirds [103], Oxford Flowers [74], Stanford Dogs [51] and Stanford Cars [31]. As seen, a higher FID score might potentially lead to relatively higher accuracy on full fine-tuning, and vice versa. This is consistent with the hypothesis claimed in §4.3.

## G  DISCUSSION

### G.1  ASSET LICENSE AND CONSENT

The majority of Visual Prompt Tuning (VPT) [49] is licensed under CC-BY-NC 4.0. Portions of [49] are available under separate license terms: google-research/task_adaptation and huggingface/transformers are licensed under Apache-2.0; Swin-Transformer [66] and ViT-pytorch [24] are licensed under MIT; and MoCo-v3 [16] and MAE [41] are licensed under CC BY 4.0. Fréchet Inception Distance (FID) [43; 91] is licensed under Apache License 2.0.

Table 10: **VTAB-1k [127]** *Natural* **per-task results for ViT-B/16 [24] pretrained on MAE [41].** Though prompt tuning shows inferior performance to full finetuning, its parameter-efficient nature makes it applicable in parameter-sensitive scenes. Same for MOCO [16] applications.

| ViT-B/16 [24] (85.8M) | VTAB-1k [127] *Natural* [7] | | | | | | | Mean |
|---|---|---|---|---|---|---|---|---|
| | CIFAR-100 | Caltech101 | DTD | Flowers102 | Pets | SVHN | Sun397 | |
| FT [47] | **24.6** | **84.2** | 56.9 | **72.7** | **74.4** | **86.6** | 15.8 | 59.31 |
| VPT [49] | 8.2 | 55.2 | **58.0** | 39.3 | 45.2 | 19.4 | **21.9** | 35.31(2) |
| - Tuned / Total (%) | 0.20 | 0.20 | 0.15 | 0.10 | 0.04 | 0.54 | 0.41 | 0.23 |

Table 11: **VTAB-1k [127]** *Specialized* **per-task results for ViT-B/16 [24] pretrained on MAE [41].**

| ViT-B/16 [24] (85.8M) | VTAB-1k [127] *Specialized* [4] | | | | Mean |
|---|---|---|---|---|---|
| | Patch Camelyon | EuroSAT | Resisc45 | Retinopathy | |
| FT [47] | **81.8** | 94.0 | **72.3** | 70.6 | **79.68** |
| VPT [49] | 77.9 | **94.9** | 45.4 | **73.6** | 72.95(2) |
| - Tuned / Total (%) | 1.06 | 1.07 | 0.15 | 0.02 | 0.57 |

## G.2 REPRODUCIBILITY

This paper is implemented in Pytorch [80]. Experiments are conducted on NVIDIA A100-80GB GPUs. For reproducibility, our full implementation shall be publicly released upon paper acceptance.

## G.3 MORE DISCUSSION ON DATASET CAPACITY

Previous works [107] demonstrate that prompt tuning in language also shows same trend that prompt tuning achieves better performance when lacking task-specific data.

Similarly in vision perspective, VPT [37] partially explore this question by claiming a consistently advanced performance to full fine-tuning across training data sizes, which is proved to be mistaken in our paper (see §4.4) when the dataset scale continue to expand. Our research undertakes a subsequent update of the claims made in the preceding research, rather than merely adopting them in their original form. Also, the results in [37] are shown on a different dataset benchmark (*i.e.*, FGVC [37]). Our paper contributes an additional piece to the puzzle of addressing gaps in common image recognition benchmarks.

## G.4 CHOOSING SINGLE TRAINING SAMPLE FOR ONE-SHOT LEARNING

We select one image per class for `train` and `val`, respectively. The choosing of the representive image for each class under the one-shot learning scene is important (see §4.4). We want to ensure that the chosen image is not an outlier [117]. Instead, it should share common features with other images from the same class. We thus apply a method called iterative testing. Specifically, we experiment with different images from the same class and evaluate our model. If the model's performance falls in a reasonable range, we state that the chosen images are representative. Also, data augmentation methods are applied, which helps to fix the variations of the image.

## G.5 SOCIAL IMPACT

This work systematically investigates several hypotheses to demystify the mechanisms behind visual prompt tuning's success. We carefully study whether its success is provided by its enhanced resilience against overfitting, flexibility in task transfer, efficient learning on small datasets, or improved optimization due to additional dimensions. Through extensive experiments on 19 diverse datasets and tasks, we reveal that surprisingly, overfitting is not the root cause of full tuning's inferior performance. Prompt tuning demonstrates better generalization when there is limited data, while full tuning catches up or even surpasses prompt tuning when more data is presented. However, prompt tuning is still a competitive approach considering its parameter-efficient nature. Overall, our paper suggests that researchers should have a clear view of the task disparities and dataset scale of down-

Table 12: **VTAB-1k [127] *Strcutured* per-task results for ViT-B/16 [24] pretrained on MAE [41].**

| ViT-B/16 [24] (85.8M) | VTAB-1k [127] *Structured* [8] | | | | | | | | Mean |
|---|---|---|---|---|---|---|---|---|---|
| | Clevr/cnt. | Clevr/dist. | DMLab | KITTI/dist. | dSprites/loc. | dSprites/ori. | SmallNORB/az. | SmallNORB/elev. | |
| FT [47] | **67.0** | **59.8** | **45.2** | **75.3** | **72.5** | **47.5** | **30.2** | **33.0** | **53.82** |
| VPT [49] | 39.0 | 40.9 | 30.6 | 53.9 | 21.0 | 12.1 | 11.0 | 14.88 | 27.91(0) |
| - Tuned / Total (%) | 0.54 | 2.11 | 1.07 | 0.54 | 0.12 | 0.55 | 2.12 | 2.11 | 1.14 |

Table 13: **VTAB-1k [127] *Natural* per-task results for ViT-B/16 [24] pretrained on MOCO [16].**

| ViT-B/16 [24] (85.8M) | VTAB-1k [127] *Natural* [7] | | | | | | | Mean |
|---|---|---|---|---|---|---|---|---|
| | CIFAR-100 | Caltech101 | DTD | Flowers102 | Pets | SVHN | Sun397 | |
| FT [47] | 57.6 | **91.0** | 64.6 | **91.6** | 79.9 | **89.8** | 29.1 | 71.95 |
| VPT [49] | **70.1** | 88.3 | **65.9** | 88.4 | **85.6** | 57.8 | **35.7** | 70.26(4) |
| - Tuned / Total (%) | 0.20 | 0.20 | 0.15 | 0.10 | 0.04 | 0.54 | 0.41 | 0.23 |

stream finetuning, and carefully select appropriate way for finetuning under "pretrain-then-finetune" paradigm.

### G.6 LIMITATIONS AND FUTURE WORK

Although we investigate several interesting and compelling doubts between full finetuning and visual prompt tuning, which are of great significance to the research community, it also comes with new challenges and unveils some intriguing questions. For example, the examination of visual prompt tuning reveals similarities to prompt tuning approaches in NLP [13; 107] and vision from different perspectives (*e.g.*, [114] discusses that prompt tuning is highly robust to label noises; [76] demonstrates how prompt-tuning enables the model to attend to context-relevant information), thereby suggesting the need for further investigation to facilitate a comprehensive and unified study. This topic can be extended to parameter-efficient methods other than prompt tuning techniques. For example, we build limited experiments on LoRa [46], showing that there is similar trend on dataset capacity when comparing with full fine-tuning (see Table 20). Another essential future direction deserving of further investigation is the the design and analysis of network's attention position and *ad-hoc* interpretability. In our paper, we follow common practice and propose network interpretability through various visualization explanation methods (*i.e.*, GradCAM, IG) at the final layer of the network. However, we highlight that visualizing the attention from cls token to other tokens [9] can be an alternative approach to understand the effect of VPT on feature. We share include different visualization position in future research. We further highlight [94] a very important work in discussing the relation between attention visual evidence and performance. During our current research, the visual evidence can hardly provide an intuitive/straightforward explanation for the performance gain, while some current works [94] pave a path for such the connection. Also, the discussed visualization approaches can be generally categorized into the *post-hoc* explanability (*i.e.*, producing explanations for trained networks by importance values [97; 126; 132; 27; 70; 95] or sensitivities of inputs [67; 52; 137]). Such approaches, however, might suffer from possible misleading [87; 88; 57; 2]. Therefore, it is worth further investigating the *ad-hoc* explanability of full finetuning and prompt tuning [81; 109] (*i.e.*, case/concept-based reasoning), particularly in decision-critical and safety-sensitive scenarios [78; 25]. The investigation of prompt tuning's *ad-hoc* explanability in research is still relatively scarce and necessitates further exploration.

Table 14: **VTAB-1k [127]** *Specialized* **per-task results for ViT-B/16 [24] pretrained on MOCO [16].**

| ViT-B/16 [24] (85.8M) | VTAB-1k [127] *Specialized* [4] | | | | Mean |
|---|---|---|---|---|---|
| | Patch Camelyon | EuroSAT | Resisc45 | Retinopathy | |
| FT [47] | **85.1** | **96.4** | **83.1** | **74.3** | 84.72 |
| VPT [49] | 83.1 | 91.0 | 81.2 | 74.0 | 82.33(0) |
| - Tuned / Total (%) | 1.06 | 1.07 | 0.15 | 0.02 | 0.57 |

Table 15: **VTAB-1k [127]** *Structured* **per-task results for ViT-B/16 [24] pretrained on MOCO [16].**

| ViT-B/16 [24] (85.8M) | VTAB-1k [127] *Structured* [8] | | | | | | | | Mean |
|---|---|---|---|---|---|---|---|---|---|
| | Clevr/cnt. | Clevr/dist. | DMLab | KITTI/dist. | dSprites/loc. | dSprites/ori. | SmallNORB/az. | SmallNORB/elev. | |
| FT [47] | **55.2** | **56.9** | **44.6** | **77.9** | **63.8** | **49.0** | **31.5** | **36.9** | 51.98 |
| VPT [49] | 48.5 | 55.8 | 37.2 | 64.6 | 52.3 | 26.5 | 19.4 | 34.8 | 42.39(0) |
| - Tuned / Total (%) | 0.54 | 2.11 | 1.07 | 0.54 | 0.12 | 0.55 | 2.12 | 2.11 | 1.14 |

Table 16: **VTAB-1k [127]** *Natural* **per-task prompt length for ViT-Base [24] pretrained on supervised ImageNet-21k.**

| ViT-B/16 [24] (85.8M) | VTAB-1k [127] *Natural* (7) | | | | | | | Mean |
|---|---|---|---|---|---|---|---|---|
| | CIFAR-100 | Caltech101 | DTD | Flowers102 | Pets | SVHN | Sun397 | |
| VPT [49] | 100 | 5 | 1 | 200 | 50 | 200 | 1 | 79.58 |

Table 17: **VTAB-1k [127]** *Specialized* **per-task prompt length for ViT-Base [24] pretrained on supervised ImageNet-21k.**

| ViT-B/16 [24] (85.8M) | VTAB-1k [127] *Specialized* [4] | | | | Mean |
|---|---|---|---|---|---|
| | Patch Camelyon | EuroSAT | Resisc45 | Retinopathy | |
| VPT [49] | 5 | 50 | 50 | 10 | 28.75 |

Table 18: **VTAB-1k [127]** *Structured* **per-task prompt length for ViT-Base [66] pretrained on supervised ImageNet-21k.**

| ViT-B/16 [24] (85.8M) | VTAB-1k [127] *Structured* [8] | | | | | | | | Mean |
|---|---|---|---|---|---|---|---|---|---|
| | Clevr/cnt. | Clevr/dist. | DMLab | KITTI/dist. | dSprites/loc. | dSprites/ori. | SmallNORB/az. | SmallNORB/elev. | |
| VPT [49] | 100 | 200 | 100 | 100 | 100 | 100 | 200 | 200 | 137.5 |

Table 19: **FGVC [49] per-task results for ViT-Base/16 [24]** pretrained on supervised ImageNet-21k.

| ViT-Base/16 [24] (85.8M) | FGVC [49] [5] | | | | | Mean |
|---|---|---|---|---|---|---|
| | CUB-200-2011 | NAbirds | Oxford Flowers | Stanford Dogs | Stanford Cars | |
| FULL [47] | 87.3 | 82.7 | 98.8 | 89.4 | **84.5** | 88.54 |
| VPT [49] | **88.5** | **84.2** | **99.0** | **90.2** | 83.6 | 89.11 (4) |
| - Tuned / Total (%) | 0.29 | 1.02 | 0.14 | 1.17 | 2.27 | 0.98 |
| FID score | 146.845 | 169.117 | 167.690 | 143.840 | 182.487 | - |

Table 20: **Cifar-100 [54] for ViT-Base [24] (bottom-up)** pretrained on supervised ImageNet-21k under the settings of [94].

| # training sample | 400 | 800 | 10000 |
|---|---|---|---|
| Full fine-tuning | 44.5% | 70.2% | 87.9% |
| LoRa [46] | 69.6% | 83.6% | 90.7% |
| Performance gap | 25.1% | 13.4% | 2.8% |

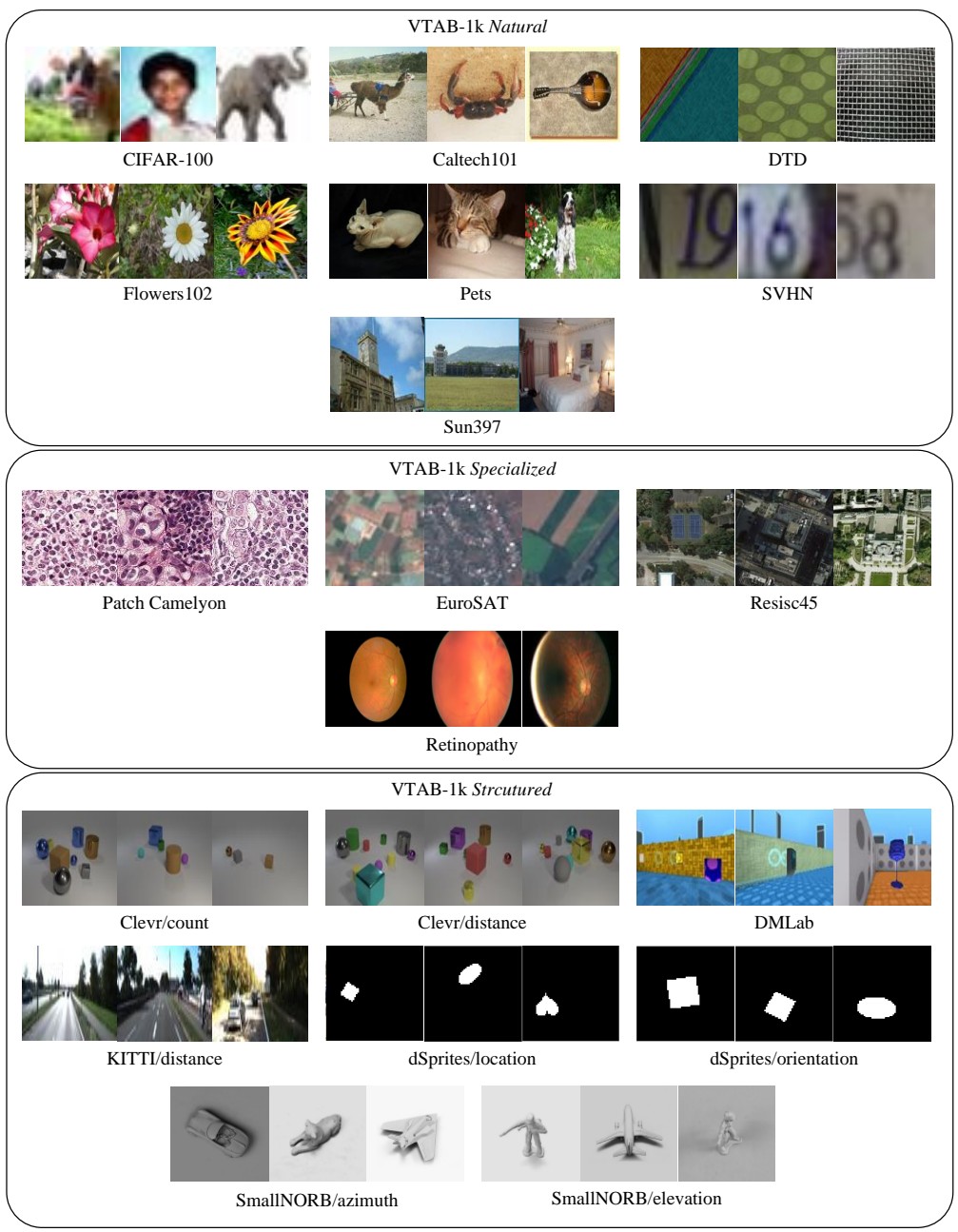

Figure 19: **Dataset examples from VTAB-1k [127]** image classification benchmark.

