# OpenReview forum: "Facing the Elephant in the Room: Visual Prompt Tuning or Full finetuning?"
_ICLR.cc/2024/Conference — ICLR 2024 poster_

### Official Review · Reviewer_5V4Y · 2023-10-22

**Soundness:** 4 excellent
**Presentation:** 3 good
**Contribution:** 3 good
**Rating:** 8
**Confidence:** 4

**Summary:**

This paper mainly focuses on visual prompt tuning. The authors conduct extensive empirical study to illustrate when and why VPT performs better than vanilla finetuning. The experiment results lead to interesting conclusion with regard to the discrepancy of task objectives and data distributions. The authors further inspect the role of solving overfitting and optimization in two kinds of methods.

**Strengths:**

1. The experiment results are sufficient.
2. The idea of explaining VPT with the different role of additional prompt parameters and original pretrained parameters is interesting.

**Weaknesses:**

1. I suggest the authors reorganize the experiment results. The current ones are confusing since the Mixed and FT-then-PT are introduced in the later part of the paper while the corresponding results are shown before.
2. I wonder if prompt size is one of the tuned hyper-parameters. If so, it would be better to present the exact number of prompts used for each setting since different prompt size indicates different learning capacity.
3. In fact the original VPT has two variants, i.e. VPT-shallow and VPT-deep. It seems VPT-deep is adopted in this paper. I wonder if the same pattern holds for VPT-shallow too.
4. It is noteworthy that VPT only performs better in half of all datasets of VTAB-1k Structured in one-shot setting as shown in Tab.4, which is different from results of other two sub-categories. This should be highlighted and illustrated in Sec.4.4.
5. The current scope of this paper is ok. But it would be much greater if the authors could expand the study to prompt tuning in language models.

**Questions:**

Please refer to the weaknesses.

---

> ### Author Response · Authors · 2023-11-17
> **Point-to-Point Response to Reviewer 5V4Y (Part I)**
>
> We thank reviewer 5V4Y for the valuable time and constructive feedback.
>
> #### **Q1 Reorganize the experiment results:**
>
> **A1:** Thank you for your valuable suggestion! We have added additional comments on "Mixed" and "FT-then-PT" to ensure that readers can easily follow the settings for future reference. These new comments offer a more logical and coherent sequence. Thank you.
>
> #### **Q2 Prompt size:**
>
> **A2:** This is an excellent question. We adhere to the default settings in VPT [ref1] for prompt length in various tasks to achieve optimal performance. For the sake of completeness, we have included an additional table in Appendix Section E, specifically Tables 16 to 18, in our revised paper. These tables are also provided below for your reference, corresponding to Natural, Specialized, and Structured tasks, respectively. Thank you.
>
> | Dataset  | CIFAR-100 | Caltech101 | DTD | Flowers102 | Pets | SVHN | Sun397 |
> | :-: | :-: | :-: | :-: | :-: | :-: | :-: | :-: |
> | Prompt Length | 100 | 5 | 1 | 200 | 50 | 200 | 1 |
>
> | Dataset  | Patch Camelyon | EuroSAT | Resisc45 | Retinopathy |
> | :-: | :-: | :-: | :-: | :-: |
> | Prompt Length | 5 | 50 | 50 | 10 |
>
> | Dataset  | Clevr/cnt. | Clevr/dist. | DMLab | KITTI/dist. | dSprites/loc. | dSprites/ori. | SmallNORB/az. | SmallNORB/elev |
> | :-: | :-: | :-: | :-: | :-: | :-: | :-: | :-: | :-: |
> | Prompt Length | 100 | 200 | 100 | 100 | 100 | 100 | 200 | 200 |
>
> #### **Q3 Experiment on VPT-shallow:**
>
> **A3:** Thank you for your careful review. Indeed, VPT suggests two versions: VPT-Shallow and VPT-Deep. While most current work [ref2-5] focuses on comparing VPT-Deep, we use VPT-Deep as our default VPT version for all experiments. However, we do observe a similar trend in VPT-Shallow regarding the findings stated in our paper. Please note that the criteria may vary, such as the performance turning point of VPT and full fine-tuning, as discussed in Sec. 4.4, since the performance of VPT-Shallow is significantly lower than that of VPT-Deep. Thank you.
>
> #### **Q4 The authors should highlight and illustrate in Sec.4.4 that VPT only performs better in half of all datasets of VTAB-1k Structured in one-shot setting:**
>
> **A4:** Thank you for the suggestion; we have added discussions to emphasize this observation. Overall, one-shot experiments favor VPT, but the results are not entirely consistent, with VPT performing better in only half of the structured cases. The accuracies for one-shot experiments are relatively low. In the revised version, we have noted that randomness could play a role in one-shot experiments, and the results may not be reliably conclusive.

---

> ### Author Response · Authors · 2023-11-17
> **Point-to-Point Response to Reviewer 5V4Y (Part II)**
>
> #### **Q5 Extension to language models:**
>
> **A5:** First, we would like to express our gratitude for your positive comments regarding the scope and research field of our paper. Second, it is important to note that in our paper (see Appendix E.4), we explicitly mention that the prompt tuning methods in vision draw significant inspiration from works in the domain of language. Consequently, there is a natural connection between these two domains, warranting further discussion.
>
> During the course of our study, we found encouragement in related studies on prompt tuning in language. For instance, [ref6] highlights a similar trend in the language domain, where prompt tuning leads to enhanced performance when task-specific data is scarce (as observed in Sec. 4.4, where substantial performance gaps arise when downstream datasets are small in size). Furthermore, we explored the concept of prompt tuning from various perspectives in both language and vision domains. Specifically, [ref7] discusses the robustness of prompt tuning in handling label noises, while [ref8] demonstrates how prompt tuning empowers the model to focus on context-relevant information.
>
> Our method delves deeper into the possible reasons and mechanisms behind the performance improvements achieved through visual prompt tuning. It also provides a thorough comparison with full fine-tuning and other competitive methods. We firmly believe that our work contributes significantly to the future exploration of intriguing topics related to prompt tuning in both vision and language.
>
> Third, it is worth noting that the overall discussion surrounding prompt tuning studies in both vision and language remains somewhat limited. To address this, we have included additional discussions on the potential extensions to language models in Appendix Sec. G.3 and G.6, ensuring a comprehensive coverage of the subject matter.
>
> Once again, we sincerely appreciate your feedback and interest in our work.
>
>
> [ref1] Visual Prompt Tuning. ECCV 2022
>
> [ref2] AdaptFormer: Adapting Vision Transformers for Scalable Visual Recognition. NeurIPS 2022
>
> [ref3] E^2VPT: An Effective and Efficient Approach for Visual Prompt Tuning. ICCV 2023
>
> [ref4] Pro-tuning: Unified Prompt Tuning for Vision Tasks. TCSVT 2023
>
> [ref5] Learning Expressive Prompting With Residuals for Vision Transformers. CVPR 2023
>
> [ref6] No more fine-tuning? an experimental evaluation of prompt tuning in code intelligence. ACM ESEC/FSE
>
> [ref7] Why Is Prompt Tuning for Vision-Language Models Robust to Noisy Labels? ICCV 2023
>
> [ref8] On The Role of Attention in prompt-tuning. ICML 2023
>
> We appreciate your thoughtful review and we hope we addressed your concerns. Please let us know if you'd like any further information. The discussed concerns are also updated in our revision paper in purple. Thank you.

---

> ### Comment · Reviewer_5V4Y · 2023-11-21
>
> Thank the authors for providing comprehensive feedback. Most of my questions have been answered and I support acceptance of the paper.
>
> The authors mention the randomness of one-shot experiments. I think it would be better to report both mean accuracy and standard deviation, which is the traditional way in few-shot learning and can better reflect the variation of the methods in my opinion.

---

> > ### Author Response · Authors · 2023-11-21
> > **Thank you for your response**
> >
> > Thank you for the prompt response. We would like to express our sincere gratitude for your constructive feedback. Your opinions have significantly elevated the quality and clarity of our paper.
> >
> > Regarding the one-shot experiments, we will report both mean and standard deviation as suggested in the revision. Once again, thank you for your valuable suggestions.
> >
> > Best,
> > Authors

---

### Official Review · Reviewer_jEi5 · 2023-10-30

**Soundness:** 2 fair
**Presentation:** 3 good
**Contribution:** 3 good
**Rating:** 8
**Confidence:** 3

**Summary:**

The authors attempt to answer the question: When and why VPT is effective. They conduct extensive experiments on 19 diverse datasets and tasks, wherein VPT outperformed FT in 16 instances (VTAB-1k [110] image classification benchmark). The model is ViT (and the authors observed the same tendency when they use Swin).

In exploring the “when” dimension, they find that VPT is preferrable when there is 1) a substantial disparity between the original and the downstream task objectives (e.g., transitioning from classification to counting), or 2) a similarity in data distributions between the two tasks (e.g., both involve natural images).
In exploring the “why” dimension, they find that VPT’s success cannot be attributed solely to overfitting and optimization considerations and that VPT preserves features and add parameters in a unique manner.

Specifically,
- VPT reaches superior performance than FT in two different scenarios: (1) when the disparity between the task objectives of the original and downstream tasks is high, or (2) when the data distributions are similar.
- The performance gap between full finetuning and prompt tuning decreases as the amount of training data increases.
- Although full finetuning generally achieves higher accuracy when rich data examples are available, the prompt tuning still reaches a competitive performance with much fewer parameters.
- In one-shot classification, prompt tuning outperforms full finetuning in many tasks and achieving substantially higher accuracy in some cases (i.e., 41.66% vs 56.94% in VTAB-1k Natural).
- Training and testing losses consistently decrease in scenarios where the task objectives are similar between the original and downstream tasks. In such cases, overfitting is not the cause of performance degradation of FT.
- The presence of additional dimensions does not significantly aid the optimization process.
- Preserving the original feature proves to be crucial for transferring to downstream tasks, as evidenced by the superior performance of VPT compared to FT-then-PT.
- Maintaining a fixed feature space in VPT may compel it to learn more valuable embeddings with the trainable prompts, as it significantly outperforms the Mixed method.
- Therefore, pretrained parameters play a pivotal role in capturing general features in transfer learning. That is, the preservation of initial
features is important for VPT’s success, but in a very sophisticated manner.

Overall, the authors find many characteristics of VPT in the context of "FT vs. VPT," although some of them might be an overstatement.

**Strengths:**

- Topic is timely, challenging, and important in machine learning community.

- The experiments are well-organized. The results are clearly shown and easy to follow.

- Error bars are provided when necessary.

- Full implementation will be publicly released for reproducibility.

- The present paper is well-written and easy to follow. Logic flow is smooth.

- Overall, the authors find many characteristics of VPT in the context of "FT vs. VPT," which contribute to the community.

**Weaknesses:**

- [Minor] The experiment is limited to Transformer-based models.

- There may be some overstatements, e.g., in Figure 4 (see Questions).

- The reason why VPT outperforms FT in scenarios characterized by distinct data distributions and high task disparity is still unclear, although the authors make a hypothesis (Section 5.3).

- The reason why FT outperforms VPT in situations involving similar tasks with varying data distributions is still unclear, although the authors make a hypothesis (Section 5.3).

**Questions:**

- [Question] The experiment is limited to Transformer-based models. Do you think all the results are also valid when we use CNNs?

- [Comment (major)] In Figure 4,
> In general, with the dataset increasing in size, the performance gap between FT and VPT becomes narrow.

I would like to see more results before the conclusion because Figure 4 shows that some of curves are not monotonic, which is counterintuitive, and that the behaviors of the curves highly depends on the data and methods.

- [Question (major)] Figure 4: Does this result include the randomness introduced by the choice of the training set samples? The broken yellow curve in the left panel looks intensely fluctuating in the small set region, and I guess this fluctuation comes from that kind of randomness.

- [Question (major)] In one-shot classification experiment (Table 3), how did the authors choose the single training sample? The performance should highly correlate with the one-shot training sample.

- [Comment] At the end of Section 4,
> These results further support our assumption that prompt tuning is more effective than full finetuning when only limited finetuning examples are available.

Doesn't this statement conflict with the Patch Camelyon result (green curves) in Figure 4?

- [Comment] Figure 5 should use log scales. Also, all figures including Figure 5 should be vector images.

- [Comment] In Section 5.2,
> Another possible explanation for why prompt tuning can achieve superior performance is that it can escape from local minima or saddle points compared to full finetuning, due to the additional dimensions introduced by the prompts.

This is clearly an overstatement (or a misuse of words). Local minima and saddle points are rigorously defined in a mathematical way. I guess what the authors mean here is if we can observe performance gain or not. However, strictly speaking, escaping form local minima or saddle points is not  a necessary nor sufficient condition of generalization in general settings. Although I understand some papers use these words interchangeably, and my comment may be a bit nitpicking, I would like to recommend that the authors modify the statement here. By the way, it will be a dramatic research if one theoretically proves that VPT in fact makes the network escape from local minima or saddle points.

- [Comment] In Section 5.3, the authors make two hypotheses. Is there any idea to validate them in an objective and qualitative manner?

---

> ### Author Response · Authors · 2023-11-17
> **Point-to-Point Response to Reviewer jEi5 (Part I)**
>
> We thank reviewer jEi5 for the valuable time and constructive feedback.
>
> #### **Q1 The experiment is limited to Transformer-based models:**
>
> **A1:** Thank you for the question. As discussed in Sec. 2, prompt tuning methods are currently available only for adapting AI foundation models to new tasks. While most of this research is focused on Transformer-based architectures, there is no proof that the findings in our paper hold for other architectures, such as CNNs. However, when discussing Transformer-based models, they have widespread applications in both vision and language, with a series of models (e.g., BERT [ref1], RoBERTa [ref2], ViT [ref3], Swin [ref4-5], PVT [ref6-7]). Our findings in this paper might provide potential contributions to the future development of such a large family of models. Thank you.
>
> #### **Q2 Overstatement discussions:**
>
> #### **Q2.1 The dataset increasing in size, the performance gap becomes narrow:**
>
> **A2.1:** Thank you for your question. We are also eager to obtain more results to substantiate our claim. However, we would like to note the following: 1. All datasets from VTAB-1k have been utilized in our experiments, comprising 19 full-size datasets. 2. Among these 19 datasets, some of them have a limited amount of data available. For instance, VTAB-1k Natural Oxford Flowers102 [ref8] contains only 1020 images for both training and validation, as stated in Sec. 4.4. We made efforts to approach their dataset limit in order to identify the performance boundary between visual prompt tuning and full fine-tuning. 3. The results and trends presented in this paper partially align with VPT [ref9]. In VPT, they extend their results to the FGVC benchmark [ref9] (see VPT Page 8, Fig. 3). Specifically, FGVC comprises 5 benchmarked Fine-Grained Visual Classification datasets, including CUB-200-2011 [ref10], NABirds [ref11], Oxford Flowers [ref12], Stanford Dogs [ref13], and Stanford Cars [ref14]. They demonstrate similar results, indicating that the performance gap between VPT and full fine-tuning narrows as dataset capacity increases, further reinforcing our claim. Thank you.
>
> #### **Q2.2 Statement conflict with the Patch Camelyon result:**
>
> **A2.2:** The Patch Camelyon dataset belongs to the "low task disparity and distinct data distributions" quadrant (Figure 3); therefore, full fine-tuning outperforms VPT. We have made it explicit that this statement is limited to the three quadrants mentioned above.
>
> #### **Q2.3 Discussion on Section 5.2:**
>
> **A2.3:** Thank you for your constructive feedback. We agree that the hypothesis could be better phrased: it is possible that the additional dimensions introduced by the prompt simply help optimize the loss further, and our experiments have shown that this hypothesis is not true. We have modified the statement to better reflect this.
>
> #### **Q2.4 It is unclear why FT outperforms VPT in situations involving similar tasks with varying data distributions, and why VPT outperforms FT in scenarios characterized by distinct data distributions and high task disparity:**
>
> **A2.4.1:** That is an excellent question. Our hypothesis is that the backbone parameters primarily capture features of the data distribution, while additional prompt parameters can contain information about the specific downstream task. When the tasks are similar but the data distribution is distinct, it becomes more crucial to learn the data distribution. This is why FT outperforms VPT in the first case. However, when both the task and the data are distinct, FT is more susceptible to overfitting, and as a result, VPT performs better when there is limited data (see Sec. 5.1). Nevertheless, it's important to note that this does not necessarily imply that VPT handles this case very effectively.
>
> #### **Q3 Randomness introduced by the choice of the training set samples:**
>
> **A3:** Thank you for your question. It's important to take note of the randomness in the left panel. Firstly, when discussing the choice of training set samples, we adhere to common practice [ref9] and employ fixed random seeds during training. Consequently, the training data used in both full fine-tuning and visual prompt tuning should remain consistent. Secondly, we also observe that randomness is rarely encountered in other datasets, occurring in only 1 out of 19 datasets. Thirdly, based on the visualization of the dataset, SVHN [ref15] is a digit classification benchmark dataset containing 600,000 32×32 RGB images of printed digits cropped from pictures of house number plates. When collecting a specific number of images (i.e., dataset capacity) for training, the distribution for each class might vary, potentially leading to the observed oscillations. Thank you.

---

> ### Author Response · Authors · 2023-11-17
> **Point-to-Point Response to Reviewer jEi5 (Part II)**
>
> #### **Q4 How to choose the single training sample:**
>
> **A4:** Thank you for your question. The selection of a representative image for each class is crucial, and we aim to ensure that it is not an outlier [ref16]. Instead, it should exhibit common features shared with other images from the same class. To achieve this, we employ a method called iterative testing. Specifically, we experiment with different images from the same class and assess our model's performance. If the model's performance falls within a reasonable range, we conclude that the selected images are representative. Additionally, we apply data augmentation methods to mitigate image variations. Further discussion on one-shot learning can be found in Appendix Sec. G.4. Thank you.
>
> #### **Q5 Figure 5 should use log scales, all figures including Figure 5 should be vector images:**
>
> **A5:** Thank you for your suggestions. We have attached the log-scale version of Figure 5 in Appendix Section A, which is Figure 9. Additionally, we have ensured that all figures are in vector image format. Once again, thank you.
>
> [ref1] BERT: Pre-training of Deep Bidirectional Transformers for Language Understanding. NAACL 2019
>
> [ref2] RoBERTa: A Robustly Optimized BERT Pretraining Approach. ICLR 2020
>
> [ref3] An Image is Worth 16x16 Words: Transformers for Image Recognition at Scale. ICLR 2021
>
> [ref4] Swin Transformer: Hierarchical Vision Transformer using Shifted Windows. ICCV 2021
>
> [ref5] Swin transformer v2: Scaling up capacity and resolution. CVPR 2022
>
> [ref6] Pyramid vision transformer: A versatile backbone for dense prediction without convolutions. ICCV 2021
>
> [ref7] Pvt v2: Improved baselines with pyramid vision transformer. CVM 2022
>
> [ref8] Automated flower classification over a large number of classes. ICIP 2008
>
> [ref9] Visual Prompt Tuning. ECCV 2022
>
> [ref10] The caltech-ucsd birds-200-2011 dataset. 2011
>
> [ref11] Building a bird recognition app and large scale dataset with citizen scientists: The fine print in fine-grained dataset collection. CVPR 2015
>
> [ref12] Automated flower classification over a large number of classes. ICIP 2008
>
> [ref13] Novel dataset for fine-grained image categorization: Stanford dog. CVPR Workshop 2011
>
> [ref14] Fine-grained car detection for visual census estimation. AAAI 2017
>
> [ref15] Competitive Multi-scale Convolution. ArXiv
>
> [ref16] One-Shot Image Classification by Learning to Restore Prototypes. AAAI 2020
>
> We appreciate your thoughtful review and we hope we addressed your concerns. Please let us know if you'd like any further information. The discussed concerns are also updated in our revision paper in orange. Thank you.

---

> > ### Comment · Reviewer_jEi5 · 2023-11-21
> >
> > Thank you for the reply and the thorough revision including detailed experimental settings (Appendix G, etc.) and potential limitations to experimental results (Section 4, etc.). They greatly improved clarification of the paper.
> >
> > I carefully read all the comments and the other reviews, too.
> > They resolved most of my concerns and corrected my misunderstandings; therefore, I raised Rate accordingly (6 to 8).
> >
> > In view of the timely, convincing contribution, I support acceptance of the paper.
> >
> > [Typo]
> > - At the end of Section 4, "not not" should be "not".

---

> > > ### Author Response · Authors · 2023-11-21
> > > **Thanks for your response**
> > >
> > > Thank you for your prompt response. We sincerely appreciate your thoughtful and comprehensive feedback on our paper. We’ll fix the typo accordingly.
> > >
> > > Best,
> > > Authors

---

### Official Review · Reviewer_8arZ · 2023-10-31

**Soundness:** 2 fair
**Presentation:** 3 good
**Contribution:** 2 fair
**Rating:** 6
**Confidence:** 4

**Summary:**

Visual prompt tuning (VPT) and full finetuning (FT) are two commonly-used techniques to adapt a pretrained vision transformer to downstream image recognition tasks. Through extensive experiments, this paper is aiming to investigate when (under which conditions) and why VPT would outperform FT:
1. For the when part, the authors find VPT is favored when:
- there’s a disparity between pre-trained task and target downstream task;
- target downstream task share similar data distribution with the pre-trained data;
- when target labeled data is limited.
2. For the why part, the authors give several hypotheses and experiments aiming to verify them including:
- both VPT and FT suffer from overfitting so VPT’s success cannot be solely attributed to less overfitting during finetuning;
- VPT’s special way of preserving original weights and adding new parameters might be the key.

**Strengths:**

1. The paper is well-written with clear high-level motivation/intuition/ideas, along with detailed subsections that introduce the low-level implementation details. It is easy to follow and understand for the general audience in the field of recognition, vision transformer or transfer learning.
2. Indeed as in the title, the problem that the paper is trying to tackle is quite important but also not fully explored yet, i.e., the elephant in the room. VPT is widely used in multiple areas as a common tool to adapt pre-trained models to downstream tasks, not only in recognition, but also in dense prediction, vision language model and even generation nowadays. However, there are very few works talking about this classic “when and why” problem such that people could understand VPT better and also use it in a better practice.
3. Some of the extensive experiments are quite convincing, especially the analysis on downstream data scale (Sec. 4.4) and hypothesis on overfitting (Sec. 5.1).
4. The hypothesis on (a) how data distribution and task similarity gonna affect VPT’s performance (Sec 4.3); (b) why VPT outperforms FT from an optimization perspective (Sec 5.2) are quite interesting. The experiments alongside are also very inspiring.

**Weaknesses:**

1. Although the hypotheses in Sec. 4.3 and Sec. 5.2 look very interesting and inspiring (as mentioned in the Strength point 4), I’m concerned the experiments are not convincing enough to support them:
- In Sec. 4.3, it’s a bit risky to draw the conclusion using only 19 data points. Adding more tasks including the detention or dense prediction ones could be better.
- In Sec. 5.2, although Fig. 7 is very intriguing, the experiment results are not able to support it, i.e., Mixed and FT-then-PT still fall behind VPT with clear margins. This also leads to a very ambiguous conclusion in the last paragraph in Sec. 5.2.
2. The comparison of GradCAM visualization in Sec. 5.4 is not very informative (the distinction is not that huge, plus more like a calibration difference?) and it’s also hard to draw any convincing conclusion from it.
3. In Tab. 1, Mixed and FT-then-PT are included without any introduction in the Sec. 4.1 alongside, which could be very confusing to the audience. Maybe remove them from Tab. 1 and make another comparison table in Sec. 5.2, where these two methods are actually introduced?
4. The number of wins notation was introduced in Tab. 3 but has already appeared in Tab. 1 and 2 before, which could confuse the audience. Also, ‘[]’ was used for both this notation and also reference. Maybe change it to ‘()’ to avoid any confusion?
5. A similar observation on data scale (Sec. 4.4) is actually also covered in the original VPT paper/experiments (Page 8, Fig. 3).

**Questions:**

It would be great if the authors could respond to the weakness points mentioned above. Thanks!

---

> ### Author Response · Authors · 2023-11-17
> **Point-to-Point Response to Reviewer 8arZ (Part I)**
>
> We thank reviewer 8arZ for the valuable time and constructive feedback.
>
> #### **Q1 Hypotheses in Sec. 4.3 and Sec. 5.2:**
>
> #### **Q1.1 Add more tasks including the detection or dense prediction:**
>
> **A1.1:** Thank you for your concern regarding the number of datasets we employed in this work.
>
> First, we would like to clarify that we formally utilized all 19 datasets from the VTAB-1k benchmark for all our experiments to maintain consistency. This approach is notably comprehensive in terms of dataset quantity, in contrast to previous studies in the field of vision prompt tuning, which limited their investigations to significantly fewer datasets. Specifically, [ref1-2] focused their analyses on single and eight datasets, respectively. The extensive utilization of our dataset not only distinguishes our work but also significantly enhances its completeness and generality.
>
> Secondly, to address your concern further, we conducted a semantic segmentation experiment on ADE20K [ref3], resulting in a corresponding FID score of 260.7. As observed, when such a higher FID is obtained, full fine-tuning leads to superior performance. Our results align with the trend reported in VPT [ref4] and TOAST [ref5] concerning semantic segmentation.
>
> Thirdly, we present additional results on the FGVC benchmark [ref4], which includes five benchmarked Fine-Grained Visual Classification datasets: CUB-200-2011 [ref6], NABirds [ref7], Oxford Flowers [ref8], Stanford Dogs [ref9], and Stanford Cars [ref10]. A higher FID score demonstrates higher accuracy in full fine-tuning, and vice versa, consistent with the hypothesis outlined in Section 4.3. We have included these additional experiments in Appendix Section G.3. Thank you for your attention.
>
>
> | Dataset  | ADE20K |
> | :-: | :-: |
> | Full fine-tuning | 36.54 |
> | VPT | 33.42 |
>
> | Dataset  | CUB-200-2011 | NABirds | Oxford Flowers | Stanford Dogs | Stanford Cars |
> | :-: | :-: | :-: | :-: | :-: | :-: |
> | Full fine-tuning | 87.3 | 82.7 | 98.8 | 89.4 | **84.5** |
> | VPT | **88.5** | **84.2** | **99.0** | **90.2** | 83.6 |
> | FID | 146.845 | 169.117 | 167.690 | 143.840 | 182.487 |
>
> #### **Q1.2 Discussion on Sec. 5.2:**
>
> **A1.2:**
> Apologies for the confusion. Figure 7 illustrates one of the potential hypotheses, but in our experiments, we have discovered that this hypothesis is **not accurate**. As you correctly noted, the results suggest that the additional dimensions do not significantly assist the model in escaping local minima. This observation was reflected in the title of Section 5.2, as well as in the last paragraph of Section 5.2. We have further enhanced the clarity of Figure 7's caption by explicitly stating that this hypothesis is not supported. Thank you.
>
> #### **Q2 Comparison on GradCAM:**
>
> **A2:**
> Thank you for the question. First, we agree that these are empirical observations based on visual evidence, while current studies are rare (see Sec. 5.4). In Appendix E.4, we mentioned that such visualization explanation methods (i.e., GradCAM, IG) might suffer from possible misinterpretation. Therefore, we emphasize a future direction in the network's ad-hoc interpretability. Additionally, we respectfully argue that the distinction in GradCAM visualizations shown in Figures 8 and 16 is relatively noticeable. These are the cases where full fine-tuning fails to yield a correct decision, but prompt tuning instead recognizes it successfully.
>
> We have also carefully considered calibration in different methods, and all results have been generated using the same standard. We further make cautious inferences based on our current findings regarding these full fine-tuning failure cases and empirically deduce that prompt tuning pays more attention to the important regions contributing to decision making. The results from different visualization methods (i.e., Integrated Gradients (IG) [ref11]) exhibit consistency. To further support our claim, we provide additional visual inspection results on different datasets (i.e., within four quadrants) in Appendix Sec. B. Thank you.
>
> #### **Q3 Confusion on the order of introducing Mixed and FT-then-PT:**
>
> **A3:** Thank you for bringing this to our attention! We have made the necessary updates to the content orders in our paper to enhance clarity. Specifically, we have included additional instructions on Mixed and FT-then-PT for future reference within the paper. Thank you.
>
> #### **Q4 Confusion on the notations:**
>
> **A4:** Thank you for your valuable suggestion! We have made the necessary adjustments to address the confusion regarding the notations (see Table 1 and 3) and the use of "[]" in all tables within our main paper and Appendix. Please refer to our revised paper for more details. Thank you.

---

> ### Author Response · Authors · 2023-11-17
> **Point-to-Point Response to Reviewer 8arZ (Part II)**
>
> #### **Q5 Similar observation on data scale:**
>
> **A5:** Thank you for your thorough review. We would like to address this concern in two aspects:
> First, we would like to address the claim made in [ref4], which partially answers this question by stating, "In contrast, VPT-deep still consistently outperforms Full across training data sizes." Our experiments in Sec. 4.4 demonstrates that this claim is untenable. Furthermore, we present evidence that full fine-tuning actually yields superior results as the dataset scale continues to expand. In our study, we take a critical approach to the claims made in the previous research, rather than simply adopting them in their original form.
> Secondly, in the case of VPT, the results are presented on a different dataset benchmark (i.e., FGVC). Our paper contributes an additional piece to the puzzle of addressing gaps in common image recognition benchmarks. To provide a comprehensive perspective, we have included an extended discussion in our paper that elucidates the relationship between our work and the findings of VPT. However, we acknowledge the need to incorporate a discussion regarding training data scale with VPT, and we have addressed this in Appendix Sec. G.3.
> Thank you once again for your valuable feedback.
>
>
> [ref1] On The Role of Attention in prompt-tuning. ICML 2023
>
> [ref2] Why Is Prompt Tuning for Vision-Language Models Robust to Noisy Labels? ICCV 2023
>
> [ref3] Scene Parsing Through ADE20K Dataset. CVPR 2017
>
> [ref4] Visual Prompt Tuning. ECCV 2022
>
> [ref5] TOAST: Transfer Learning via Attention Steering. ArXiv 2023
>
> [ref6] The caltech-ucsd birds-200-2011 dataset. 2011
>
> [ref7] Building a bird recognition app and large scale dataset with citizen scientists: The fine print in fine-grained dataset collection. CVPR 2015
>
> [ref8] Automated flower classification over a large number of classes. ICIP 2008
>
> [ref9] Novel dataset for fine-grained image categorization: Stanford dog. CVPR Workshop 2011
>
> [ref10] Fine-grained car detection for visual census estimation. AAAI 2017
>
> [ref11] Axiomatic Attribution for Deep Networks, ICML 2017
>
> We appreciate your thoughtful review and we hope we addressed your concerns. Please let us know if you'd like any further information. The discussed concerns are also updated in our revision paper in blue. Thank you.

---

> > ### Comment · Reviewer_8arZ · 2023-11-21
> > **Thank you for your response!**
> >
> > Thanks to the authors for their detailed point-to-point response. It resolved my concerns including weakness points 1(a), 3, 4, and 5.
> >
> > However, I'm still concerned in terms of point 1(b) and 2, i.e, the Sections. 5.2~5.4, especially the optimization hypotheses. If I understand correctly, the authors came up with this hypothesis (which is not mentioned in previous published works) and proved it wrong with two VPT variants (which actually performs worse than the original VPT), and then listed it as one of the contribution? I do not deny that it's an interesting observation (as I mentioned in the strength point), but I'm still concerned whether this observation is significant enough to be put in the main paper as one contribution. Similarly, I still stay with my original comments in terms of the GradCAM visualization, which I think it's not convincing enough to validate the effect of VPT.
> >
> > Based on the reasons above, I would change my rating to 6.

---

> > > ### Author Response · Authors · 2023-11-22
> > > **Thank you for the response**
> > >
> > > Thank you for your positive response. We want to express our sincere gratitude for your constructive feedback.
> > >
> > > Regarding the optimization hypothesis, we fully understand your point about the value of the experiments that prove the hypothesis to not hold. The proposed experiment also supports our second hypothesis: the mere increase of parameters is inadequate, fixing part of the parameters to preserve learned features is one of the reasons that VPT could work. The significance of this experiment lies in both ruling out one hypothesis and evidence for another. Thank you for your suggestion, and we will further revise the paper to emphasize this point.
> > >
> > > Regarding the visualization explanation, we do agree that these can not be a strong support to validate the effectiveness of VPT. Our intention was for these visualizations to function as supplementary visual evidence, offering a point of reference for fellow researchers. It is important to note that contemporary visualization methodologies in the context of visual prompt tuning are still relatively scarce. As stated in Sec.G.6, our primary objective is to develop explainability in our research to further unbox the relationship between visual evidence and performance. We’ll revise accordingly to make it more clear.
> > >
> > > Once again, thank you for your valuable suggestions. Your opinions have significantly increased the quality and clarity of our paper.

---

### Official Review · Reviewer_2Y28 · 2023-11-02

**Soundness:** 3 good
**Presentation:** 3 good
**Contribution:** 4 excellent
**Rating:** 8
**Confidence:** 4

**Summary:**

The paper provides a deep analysis of visual prompt tuning (VPT), a popular transfer learning method for vision tasks. Specifically, the paper  discusses when VPT is more favorable than fully fine-tuning and give some insights on why VPT has higher performance than fully fine-tuning. The paper finds that when the data disparity between pre-training and downstream tasks is small, or the task disparity is large, VPT normally has higher performance. The paper demonstrates that when task disparity is large, VPT is usually better because fully fine-tuning is more prone to over-fitting, and when data disparity is small, VPT is usually better because it preserves more information from pre-training by freezing the parameters. The paper provides insights and guidance on when and why to choose VPT over fully fine-tuning.

**Strengths:**

The paper discusses an interesting and important subject, and gives sensible and valuable insights on the topic. The whole analysis is thorough enough and the conclusions seem credible. The paper is overall well written and easy to follow.

**Weaknesses:**

1. In Section 5.4, the authors show that the GradCAM maps of VPT is more interpretable and more focused on task-relevant regions than fully fine-tuning. It seems the examples are all from natural image classification. It would be helpful to compare the attention on four quadrants of transfer learning to see the difference when data/task disparity varies and how it is connected to the conclusion in Section 4.

2. The visualization of attention in Section 5.4 is based on GradCAM. Have the authors tried visualizing the attention from cls token to other tokens, as in previous work in the ViT literature (e.g., [1])? On the other hand, a previous paper [2] also discusses the relation between transfer learning and attention. They find that current transfer learning methods tend to have noisy attention and fixing the attention can boost the transfer learning performance. It would be helpful to add some discussion on the relation between this work and [2].

3. This paper is focused on VPT. Since other transfer learning methods such as LoRA [3] may have more favorable performances on various transfer learning tasks like VTAB-1k, readers may wonder if these methods also have similar properties. It would be helpful to see if the same conclusions also hold for other transfer learning methods.

[1] Caron, Mathilde, et al. "Emerging properties in self-supervised vision transformers." Proceedings of the IEEE/CVF international conference on computer vision. 2021.
[2] Baifeng Shi, Siyu Gai, Trevor Darrell, and Xin Wang. Refocusing is key to transfer learning. arXiv preprint arXiv:2305.15542, 2023
[3] Hu, Edward J., et al. "Lora: Low-rank adaptation of large language models." arXiv preprint arXiv:2106.09685 (2021).

**Questions:**

See Weakness.

---

> ### Author Response · Authors · 2023-11-17
> **Point-to-Point Response to Reviewer 2Y28**
>
> We thank reviewer 2Y28 for the valuable time and constructive feedback.
>
> #### **Q1 Compare the attention on four quadrants of transfer learning:**
>
> **A1:** This is an excellent point! The examples currently presented in Figure 8 and Appendix Sec. B are drawn from a single dataset in Natural, which might introduce bias in our understanding of the effect of VPT on features. However, during our experiments, we consistently observed visual evidence across different datasets. We have included attention visualization (i.e., GradCAM) for the other three quadrants of transfer learning in Appendix Sec. B. The visual evidence suggests that in the 1st-3rd quadrants, visual prompt tuning generally results in a denser and more accurate focus on desired positions compared to full fine-tuning. For datasets falling into the 4th quadrant, our results reveal noticeable improvements when subjected to full fine-tuning. Thank you.
>
> #### **Q2.1 Visualizing the attention from cls token to other tokens:**
>
> **A2.1:** Thank you for your question. Currently, we adhere to common practices [ref1-3] and visualize the final layer of the network. However, it's worth noting that an alternative approach to understanding the impact of VPT on features is to visualize the attention from the cls token to other tokens [ref4]. We have also attempted to visualize the patch embeddings from the transformer, and these visualizations have consistently yielded meaningful insights. Given that the cls token is explicitly designed for classification, we find it reasonable to visualize the attention from the cls token to other tokens. We plan to delve deeper into these results in our future studies. Thank you.
>
> #### **Q2.2 Discussion the relation with [ref5]:**
>
> **A2.2:**
> Thank you for the reference! We consider [ref5] to be a highly significant work, particularly in the context of the discussion regarding the relationship between attention, visual evidence, and performance. In current research, it is often the case that visual evidence cannot offer an intuitive or straightforward explanation for performance improvement, but [ref5] provides a pathway to establish such a connection. We believe this topic is exceptionally intriguing and can substantially enhance our reassessment in Section 5.4. We have included a discussion on this in Appendix G.6 as part of our future work. Thank you.
>
>
> #### **Q3 Generalization to other transfer learning methods:**
>
> **A3:** Thank you for bringing this to our attention. We have observed that [ref5] presents a comparison of several baselines for transfer learning, including linear, full fine-tuning, VPT, and LoRa [ref6], which yield valuable insights. In this paper, our primary focus is on visual prompt tuning, an area that remains largely unexplored and has not been systematically studied. We narrow our scope to conduct a comprehensive and systematic investigation.
>
> When discussing LoRa, it introduces low-rank metrics and exclusively tunes these metrics while keeping other components frozen. While there are certain similarities between these two methods (namely, both can be categorized as parameter-efficient approaches [ref7-8]), their distinctions may lead to ambiguous results. We contend that our paper takes a substantial step toward a deeper understanding of prompt tuning. In the future, we naturally plan to expand our study to include LoRa and other parameter-efficient techniques.
>
> During the rebuttal phase, we managed to provide some results on LoRa (specifically, we varied the dataset scale on CIFAR-100 from 400 to 10,000 training samples for five runs and compared LoRa with the Full fine-tuning method) in the table below (following the code provided by TOAST [ref5], utilizing the ViT-B-in21k (bottom-up) checkpoint). As depicted, as the dataset scale increases, the gap between LoRa and full fine-tuning gets narrow significantly, aligning with our observations.
>
> We have included further discussion in Appendix Sec. G.6. Thank you for your attention.
>
>
> | # training sample  | 400 | 800 | 10000 |
> | :-: | :-: | :-: | :-: |
> | Full fine-tuning | 44.47 | 70.2 | 87.9  |
> | LoRa | 69.6 | 83.6 | 90.7  |
> | Performance gap | 25.1 | 13.4 | 2.8 |
>
> [ref1] Grad-cam: Visual explanations from deep networks via gradient-based localization. CVPR 2017
>
> [ref2] Adapting grad-cam for embedding networks. WACV 2020
>
> [ref3] Transformer interpretability beyond attention visualization. CVPR 2021
>
> [ref4] Emerging properties in self-supervised vision transformers. ICCV 2021
>
> [ref5] TOAST: Transfer Learning via Attention Steering. ArXiv 2023
>
> [ref6] Lora: Low-rank adaptation of large language models.
>
> [ref7] Visual Prompt Tuning. ECCV 2022
>
> [ref8] E2VPT: An Effective and Efficient Approach for Visual Prompt Tuning. ICCV 2023
>
> We appreciate your thoughtful review and we hope we addressed your concerns. Please let us know if you'd like any further information. The discussed concerns are updated in our revision paper in red. Many thanks!

---

> > ### Comment · Reviewer_2Y28 · 2023-11-21
> > **Thank you for the response**
> >
> > All of my concerns are clearly addressed. I thereby increase my score to 8.

---

> > > ### Author Response · Authors · 2023-11-21
> > > **Thank you for your response**
> > >
> > > Thank you for your prompt response. We are genuinely grateful for your thoughtful feedback. We are immensely appreciative of the discussions on the visualization and generalization, as they clearly illuminate the future direction of our work.
> > >
> > > Best,
> > >
> > > Authors

---

### Author Response · Authors · 2023-11-17
**Summary of Revisions**

Dear Reviewers:

Thank you so much for your careful review and suggestive comments. We have revised our paper according to your comments. The major changes are as follows:

1. We’ve supplemented experiments and discussion to include LoRa in Appendix Sec. G.6, by Reviewer 2Y28’s suggestion.
We’ve added additional experiments on segmentation and FGVC image classification benchmark in Appendix Sec. G.3, suggested by Reviewer 8arZ.

2. We’ve reported the additional attention visualization on other three quadrants in Appendix Sec. B, as suggested by Reviewer 2Y28, 8arZ and jEi5.

3. We’ve reported the per-task prompt length in Appendix Sec. E Table 16-18, according to Reviewer 5V4Y.

4. We’ve proposed additional discussion on our proposed statements, as suggested by 2Y28, 8arZ and jEi5.

5. We’ve introduced future extensions and discussion to the language domain in Appendix Sec. G.6, as suggested by Reviewer 8arZ and 5V4Y.

Sincerely yours,

Authors.

---

### Author Response · Authors · 2023-11-21
**Seeking an open dialogue**

Dear Reviewers,

We sincerely appreciate the time and effort you've devoted to reviewing our work. We understand that your schedule may be quite busy. As the authors-reviewer discussion phase draws to a close, we kindly request your attention to our responses. Our aim is to gain insights into whether our responses effectively address your concerns and to ascertain if there are any additional questions or points you would like to discuss.

We look forward to the opportunity for further discussion with you. Thank you for your thoughtful consideration.

Best regards,
The Authors

---

### Meta-Review · Area_Chair_zzV3 · 2023-12-10

**Metareview:**

This paper offers an insightful deep analysis of visual prompt tuning (VPT), a new paradigm in the realm of transfer learning for vision tasks. It has undergone a thorough review process by four experts in this field.

The reviewers have unanimously recognized the paper's strengths, particularly in discussing the conditions under which VPT outperforms full fine-tuning. The paper successfully sheds light on why VPT demonstrates higher performance in certain scenarios, providing multiple valuable insights that contribute meaningfully to the field.

Reflecting on the positive feedback from the reviewers, it is my pleasure to recommend this paper for acceptance at ICLR 2024. However, alongside their commendations, the reviewers have also identified several minor aspects that require further attention. The authors are therefore encouraged to address these concerns meticulously in the final camera-ready version of the paper. Congratulations again to the authors on the acceptance of their paper.

**Justification For Why Not Higher Score:**

The authors uncover many characteristics of VPT in the context of "FT vs. VPT".

**Justification For Why Not Lower Score:**

The experiments are not convincing enough to support the hypotheses in Sec. 4.3 and Sec. 5.2.

---

### Decision · Program_Chairs · 2024-01-16

Accept (poster)